# Exogenous application of hormones and chemicals to mitigate heat stress in strawberry under tropical summer conditions

**Najra-Tan-Nayeem Salwa**[☉], **Sadia Shabnam Swarna, Md. Masudur Rahman, Sharifunnessa Moonmoon**[iD][☉]*

Department of Crop Botany and Tea Production Technology, Faculty of Agriculture, Sylhet Agricultural University, Sylhet, Bangladesh

[☉] These authors contributed equally to this work.
* moondj311@yahoo.com

## Abstract

Resilience of Strawberries in tropical and subtropical climates, including Bangladesh, faces severe challenges due to rising summer temperatures and sporadic heatwaves exacerbated by climate change. The experiment evaluated the physiological and biochemical responses of three Strawberry genotypes of which G1 (RABI-3) was heat tolerant and G2 (BARI Strawberry-2) and G3 (BARI Strawberry-3) were heat susceptible and assessed the effectiveness of five exogenous substances like 5% Molasses (T1), 10 ppm Melatonin (T2), 5% Kaolin (T3), 10 mM $CaCl_2$ (T4) and 5 ppm Abscisic acid (T5) in mitigating heat induced damage. The experiment was conducted in pots under field conditions following CRD (Completely Randomized Design) set was repeated thrice. Exogenous treatments were applied as foliar spray at five days intervals for three months in comparison to the control. The hourly temperature and relative humidity (%RH) data was recorded using digital data logger. Leaf fluorescence parameters, temperature differentials, photosystem II efficiency and leaf attributes such as thickness and greenness were analyzed. The plant data was recorded three time viz. at the end of March, at the end of April and at the end of May using a handheld fluorometer (Multispeq V.02). The result showed that the heat tolerant genotype (G1) consistently maintained higher photosynthetic activity and chlorophyll content, while the susceptible genotype (G3) showed significant improvement under treatments. All the exogenous substances possess the ability to help the plant to mitigate the effects of heat stress and support photosynthetic activity. The organic spray like kaolin and molasses effectively reduced internal leaf temperature by developing surface coating, while ABA enhanced internal physiological regulation. The results suggest that the integration of heat tolerant genotypes with suitable treatments can enhance Strawberry resilience to heat stress, which provides a practical alternative for sustainable Strawberry cultivation in tropical regions.

**Data availability statement:** The data underlying the results presented in the study are available as underlined supplementary file in the submission.

**Funding:** This work was supported by the Special research grant (SRG), Ministry of Science and Technology, Government of The People's Republic of Bangladesh (grant number: SRG – BS – 307; 2023-24)" https://grant.most.gov.bd/en/services/most/special-research-grant received by corresponding author S. Moonmoon. The funders had no role in study design, data collection and analysis, decision to publish, or preparation of the manuscript.

**Competing interests:** No authors have competing interest.

## Introduction

Strawberry is a dicotyledonous perennial herb belongs to genus *Fragaria* of family Rosaceae, grown throughout the world for their sweet fragrance, taste, nutritional, and medicinal value [1, 2]. Globally, Strawberry production exceeds 9.1 million metric tons, largely driven by its' rich content of phytonutrient, antioxidant and poly-phenolic chemicals which can reduce chronic disease like cancer, heart disease etc. [3,4]. In fact, Strawberries contain more vitamin C than lemon [5]. While it typically cultivated as a summer crop in temperate region [6], in tropical and subtropical regions like Bangladesh, Strawberry is grown as a winter crop [7], yielding over 789 metric tons annually. In Bangladesh, the demand of strawberry is rapidly increasing. However, Strawberry cultivation faces significant climatic challenges. The optimal temperature range for Strawberry growth is 20–26 ℃ during day and 12–16 ℃ at night. This condition is only prevails in winter in Bangladesh thus strawberries cultivated in Bangladesh at winter [8]. The country lacks improved multiplication technique and heavily relies on the conventional easiest vegetative propagation (by runner, sucker) [9,10]. Vegetative preparation requires a mother plant; thus, farmers kept the plants till September of the next year after ending the winter season. Consequently, these mother plants must endure the tropical summer, during which temperatures regularly exceed 26℃, significantly impairing plant growth [11]. Nonetheless, the heat wave develops at the mid and late summer raise the temperature and worsen the situation. Extended exposure to heat stress results in visible symptoms like leaf wilting and burn and a prolonged heat stress may lead to mother plant mortality. To make the situation worse climate change impacts heavily. The changed climate destroyed the seasonal and weather pattern; thus, it became really hard for the farmers what to expect when. High temperatures and unpredictable heat waves are now among the main constraints to Strawberry production in tropical and subtropical climates. Despite this growing concern, limited research has focused on alleviating heat stress in Strawberries under tropical conditions. Despite the promising results of exogenous treatments in other crops and controlled conditions, little is known to suggest their effectiveness in Strawberry cultivation in tropical climates, like Bangladesh. In addition, the comparative efficacy of different hormone-based treatments and the effects of genotypic variation in the effectiveness of treatment remains largely unexplored. Further, there is a need to determine whether these treatments can sustain plant health and photosynthetic performance under prolonged periods of extreme environmental heat.

Several exogenous substances have showed potential in reducing the effects of heat stress in plants. For instance, the application of molasses in strawberry plants under heat stress conditions has been shown to have a significant effect on plant physiology and therefore has a potential tool for improving heat tolerance. According to Gulen and Eris [12], molasses increases leaf relative water content (LRWC) and assists the plant in having good turgidity for the management of water loss under high temperatures. Moreover, Kesici and colleagues, [13] also revealed that molasses increases the chlorophyll content and vigor, which is crucial for the plants to continue photosynthesis during hot and unfavorable temperatures. Melatonin is another effective compound known for its' role in enhancing the antioxidant activity of strawberries,

particularly under heat stress, due to the involvement in oxidative stress regulation, protecting fruit quality and nutrient content. Wang and colleagues, [14] confirmed that melatonin regulates the photos system electron transport in tall fescue under heat stress. The increased number of reaction centers weakens the electron transfer efficiency at the receptor and donor sides of Photosystem II (PSII), enhancing the rate of plant survival under heat shock treatment. In general, melatonin is great for reducing heat stress practically on different crops through increasing antioxidant activity, improving plant photosynthesis process and increasing the general plant condition. Melatonin application at varying concentrations (1, 5, and 10 ppm) has been reported to enhance yield, fruit quality, and antioxidant levels in multiple strawberry cultivars under both greenhouse and open-field conditions [15]. Kaolin, which is an organic mineral compound high in kaolinite, has a long history of use in reducing heat stress and drought stress on a wide variety of crops. Kaolin, when applied as a foliar spray, produces a film of white, reflective particles covering a leaf surface that reflects infrared (IR), photosynthetically active radiation (PAR), and ultraviolet (UV) radiation [16–18]. This property is important for use under heat stress since it will limit the absorption of IR by the canopy and cool the leaf. The cooling properties of kaolin were apparent with the kaolin-treated (T3) plants where the internal leaf temperature was lower than for other treatments with the surface readings slightly higher with the toxic coating from the kaolin spray. Therefore, kaolin is thought to work as an antitranspirant to avoid water loss and improve water use efficiency [19,20]. This may explain the improved SPAD values, photosystem II efficiency (Fv/Fm), and maintenance of photochemical activity ($\varphi$II, qL) observed in plants treated with kaolin in our study. The protective coverage by the kaolin treatment probably decreased transpiration water loss, decreased light activated stress, resulting in maintaining leaf structure and function at high temperatures. Kaolin particle film, at rates like 3% and 6%, reduced light and heat stress and improved quality of nut and kernel in Persian walnut [21]. Calcium chloride ($CaCl_2$) has also proven effective in enhancing plant heat tolerance. According to Tan and colleagues [22], heat tolerance improved in tobacco due to $CaCl_2$ treatment in terms of stomatal conductance and higher net photosynthetic rate, stabilizes the PS II reaction center and lower Reactive Oxygen Species (ROS) accumulation through increased antioxidant enzyme activity. There was further indication of enhanced heat stress resistance by indicated improvement levels of HSP70. Foliar application of $CaCl_2$ at concentrations of 5, 10 and 20 mM improved growth, photosynthesis, and antioxidant responses in Zoysia japonica under drought stress [23]. The exogenous application of abscisic acid (ABA) has been identified as an effective method for enhancing heat stress tolerance in strawberry plants, providing multiple interconnected advantages for both physiological and morphological adaptations. Notably, Tao and colleagues [24] found that ABA significantly improves the development of root and water absorption, helping the plants not to wilt and tolerate high temperatures. Additionally, the work of Islam and colleagues [25] highlighted that ABA promotes the accumulation of proline, an osmoprotectant that combined with an upregulated antioxidant defending system reduces the oxidative harm in the plant. Tao and colleagues [24] highlighted another feature of ABA, regulating stomatal behavior, specifically by minimizing stomatal opening, which reduces water loss. This ability to conserve moisture and ensure they remain hydrated during prolonged heat exposure.

Besides the application of these protective compounds, it is also essential to identify which genotype can cope up with high temperature. Understanding the genotypic variation in response to heat stress and the effectiveness of chemical treatment is vital for developing heat resilient Strawberry cultivation in tropical regions. Therefore, this present study was conducted to find out the suitable heat-tolerant Strawberry genotypes for summer cultivation and to investigate the impact of hormonal treatments on strawberry genotypes under heat stress conditions.

## Materials and methods

### 2.1. Study location, duration, work permission and ethical considerations

The study was conducted in the Department of Crop Botany and Tea production technology field laboratory, Sylhet Agricultural University, Sylhet-3100, Bangladesh, from October-November of 2023 to April - May of 2024. No such body is existed at both institutional and country level and no such permission is needed or applicable for conduction of any research work or experiments regarding **ethical considerations of** plant culture and growth.

## 2.2. Genotype collection

Six genotypes of strawberries were collected from different locations in Bangladesh in October-November 2023. Among these genotypes, three genotypes of one RABI-3, which is heat tolerant (G1) and two heat susceptible, BARI Strawberry-2 (G2) and BARI Strawberry-3 (G3) were selected for the experiment.

## 2.3. Plant multiplication

The collected genotypes were morpho-physiologically evaluated and reared in the pot concomitantly. The mother plant grown in the pot produces runner in case of some genotypes and sucker in rest others. The runner was provided with new pots containing rotting media for obtaining new plant. The sucker was separated from the mother plant periodically and grown in the separate pot having growing media. The full gown seedlings were transplanted in the main experimental field.

## 2.4. Pot preparation and plant establishment

Black colored buckets with an 8 L capacity were used as the growing pot with perforated bottoms for drainage. Each pot was filled with a 2 inches base layer of sand and pebbles followed by 5 kg of soil mixed thoroughly with following recommendation. 250g vermicompost, 1.67 g TSP, 1.83 g MoP and 0.08 g $MgSO_4$ as per the BARC BARC (2012) guidelines. After the pot became ready, seedling produced in the polybag was planted in the pot. Three selected genotypes (G1, G2 and G3) were taken and 9 plants from each genotype were planted in the 9 different plot (Fig 1).

## 2.5. Experimental design and treatment application

The experiment for heat stress evaluation the selected genotypes and investigating mitigation strategy were laid in the field placing the planted pot following the Completely Randomized Design CRD model [26] with three replications (Fig 2). In each replication, there was 3 plants. Five exogenous chemicals consisting of organic, inorganic and PGRs (Plant Growth Regulator) hormone were selected (Molasses-[Aarong –BARC, BD], Melatonin-[UT enterprise Co. Ltd; China], Kaolin-[UT enterprise Co.Ltd; China], $CaCl_2$-[UT enterprise Co.Ltd; China] and Abscisic acid-[Nanjing Huaxi Chemicals; China] for the mitigation of impacts of heat stress on the plants in comparison to control (Water spray) (Fig 3). All the reagents were stored at 4° C.

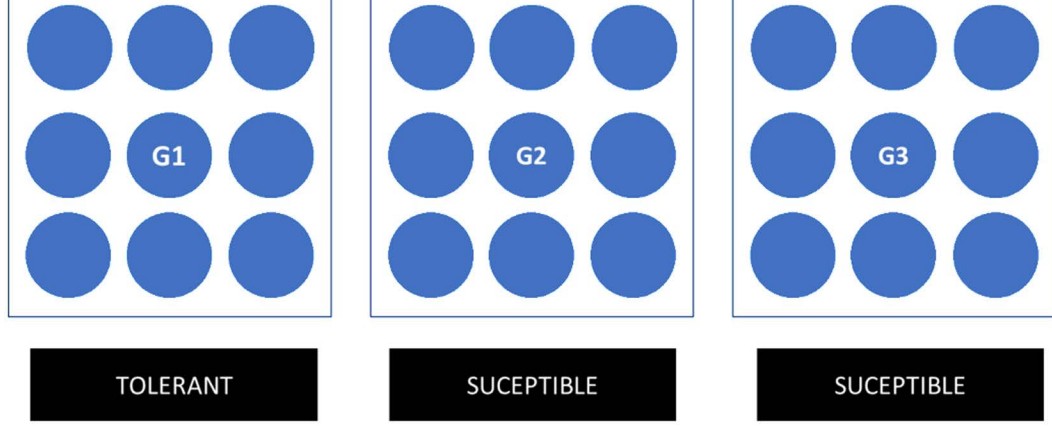

**Fig 1. selected genotypes and plants for the pot experiment.**

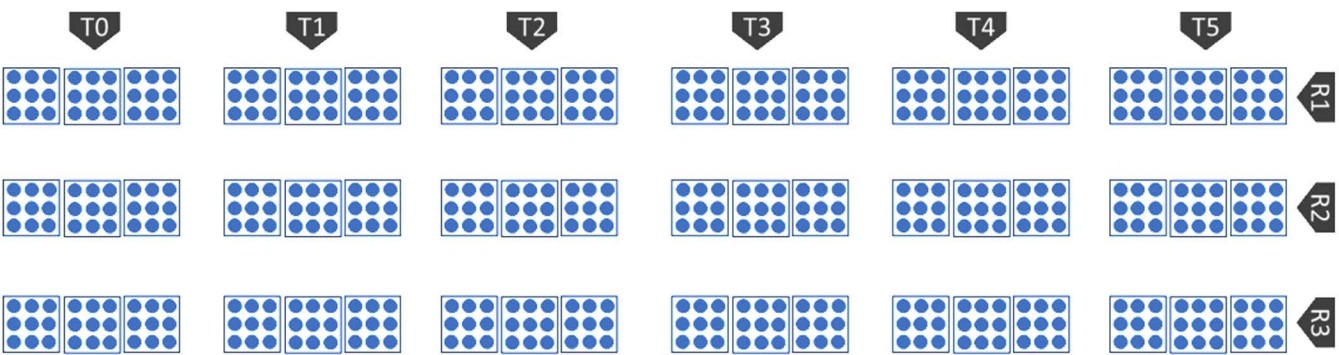

**Fig 2. experimental layout for pot culture.**

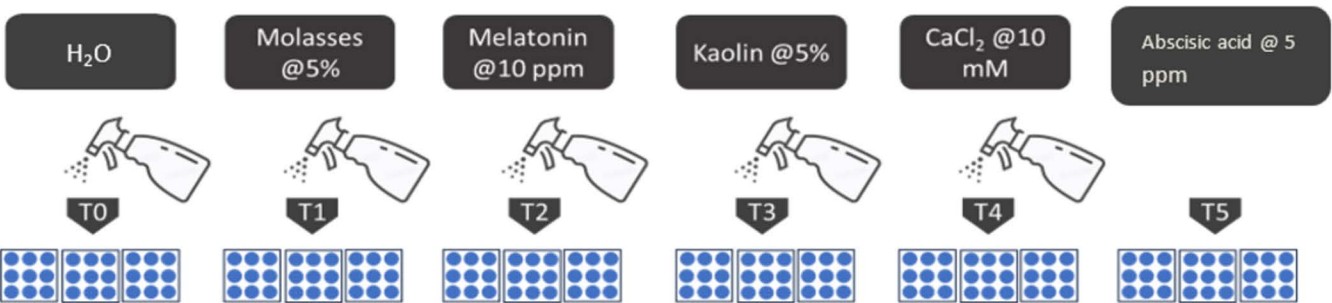

**Fig 3. selection of the exogenous spray for the experiment.**

At the first week of March when night temperature raised and day becomes hotter the application of the exogenous chemicals was started. Organic molasses was purchased from the local market and diluted to 5% Molasses solution (T1) and stored in the freeze for the application. Likewise, 10 ppm Melatonin (T2), 5% Kaolin (T3) and 10 ppm $CaCl_2$ (T4) and 5 ppm Abscisic acid (T5) were prepared and stored in the freezer for the periodic application. The spray solution was brought to room temperature after taking out from the freezer and applied at five (5) days intervals to the specified experimental units till plants became fully soaked, whereas the control plants were sprayed with distilled water at the same manner. Other operations like irrigation water application, recommended fertilizer application, pesticides and fungicides etc. application were the same for all the treatments. The spraying was continued till the end of May and stopped after collecting the final data.

## 2.6. Environmental monitoring

Meteorological data (Temperature and Relative Humidity) was collected by placing digital data loggers at the experimental site from the first week of March to end of the May to record the heat stress. The data logger was well protected from the outer environment but allowed record temperature and Relative Humidity (%RH) data from the environment without obstacle. Data were collected using the mobile app and then transferred to Microsoft excel to process and visualize.

## 2.7. Plant fluorescent data collection

Plant fluorescent response recording is a state-of-art perfect strategy to assess the plant's physical condition under heat stress and the response under the treatments. To collect the fluorescent data a handheld fluorometer (Multispeq V.2) [27]

made in USA was used. Fluorescent data were collected first half of the day of three times viz. th March, 30th April and in 30th May to get contrast insights about plant response under different intensities of heat stress. Data were recorded in the first part of the day. The fluorometer submitted the data to the server and downloaded it later on for further processing. The following parameters were recorded: Leaf surface and internal temperature, Leaf thickness and greenness (SPAD), Chlorophyll fluorescence (Fv/Fm, LEF, φII, φNO, φNPQ and qL), steady-state proton flux (vH) and proton conductivity (gH).

### 2.8. Data analysis

In the study, treatment and genotype were considered as fixed effects, and replication as a random effect within the linear mixed model (LMM) framework. All the recorded data were transferred in Microsoft Excel and data curating, sorting, arranging, processing, visualization etc., were performed. To generate clear summaries for visualization, two separate pivot tables were created from the raw data for each parameter, one summarizing data by genotype for overall treatment effects and the other summarizing data by treatment. Data was then loaded in the R software [28] and RStudio (Version: 2025.05.0 + 496) to carry out all kinds of statistical analysis. Linear mixed modelling was used to perform ANOVA in the light-dependent parameters. The light-independent parameters were analyzed by performing two-way ANOVA. The mean of the significant parameters was separated using the fisher LSD test [26] at 0.05 level of p-value. The "tidyverse" package was used for the calculation of mean and sem. The data was visualized in graph form using the "ggplot2" package along with others. Pearson's correlation test was implied in the parameters using the "metan" package along with performing the significant test.

## Results

### 3.1. Environmental data

The data were recorded every 10-minute interval for the entire period (March, April and May) with digital data logger. The daily maximum, minimum and average were calculated for the temperature and relative humidity (% RH).

**3.1.1. Maximum, minimum and average daily air temperature.** The presented graph (Fig 4) of maximum, minimum and average daily air temperature shows the trend of air temperature during the study period. In March the day and night temperatures started to rise. By the end of the month, the day temperature exceeded 35 ℃ while the night temperature exceeded 20 ℃. In that month the average day temperature was 35.5 ℃ while the average night temperature of the month was 25.1 ℃ with a total average of 18.4 ℃. In April, the temperature fluctuated more and days became hotter. The day temperature was always higher than 35 ℃ and exceeded 40 ℃ deliberately. In that month the average maximum temperature was 39.4 ℃ and the average minimum temperature was 22.88 ℃ with a total average of 31.1 ℃. The May was hottest month among the three with periodic heat-wave. The day temperature was always around 40 ℃ with a warm night around 25 ℃. The average of maximum temperature was 41 ℃ and average of minimum temperature was 24.8 ℃ with a daily average of 32.9 ℃.

**3.1.2 Maximum, minimum and average daily relative humidity (%RH).** March was relatively dry, with relative humidity ranging from an average minimum of 49.5% to a maximum of 87% and a mean daily value of 68% (Fig 5). The April was a little bit of wetter with rain but May is wetter. The maximum average relative humidity (%RH) was 87 and the minimum average of 51% with a daily average of 69%. In May, first part of the month was drier but later part was extremely humid with maximum average of 89% and minimum average of 64% with daily average of 76%.

### 3.2. Leaf temperature of the strawberry plant

The ambient temperature of the leaf surface (Fig 6), leaf internal temperature (Fig 7) and the differential temperature (Fig 8) is an important parameter to describe leafy plant condition under the heat stress. In every figure, panel

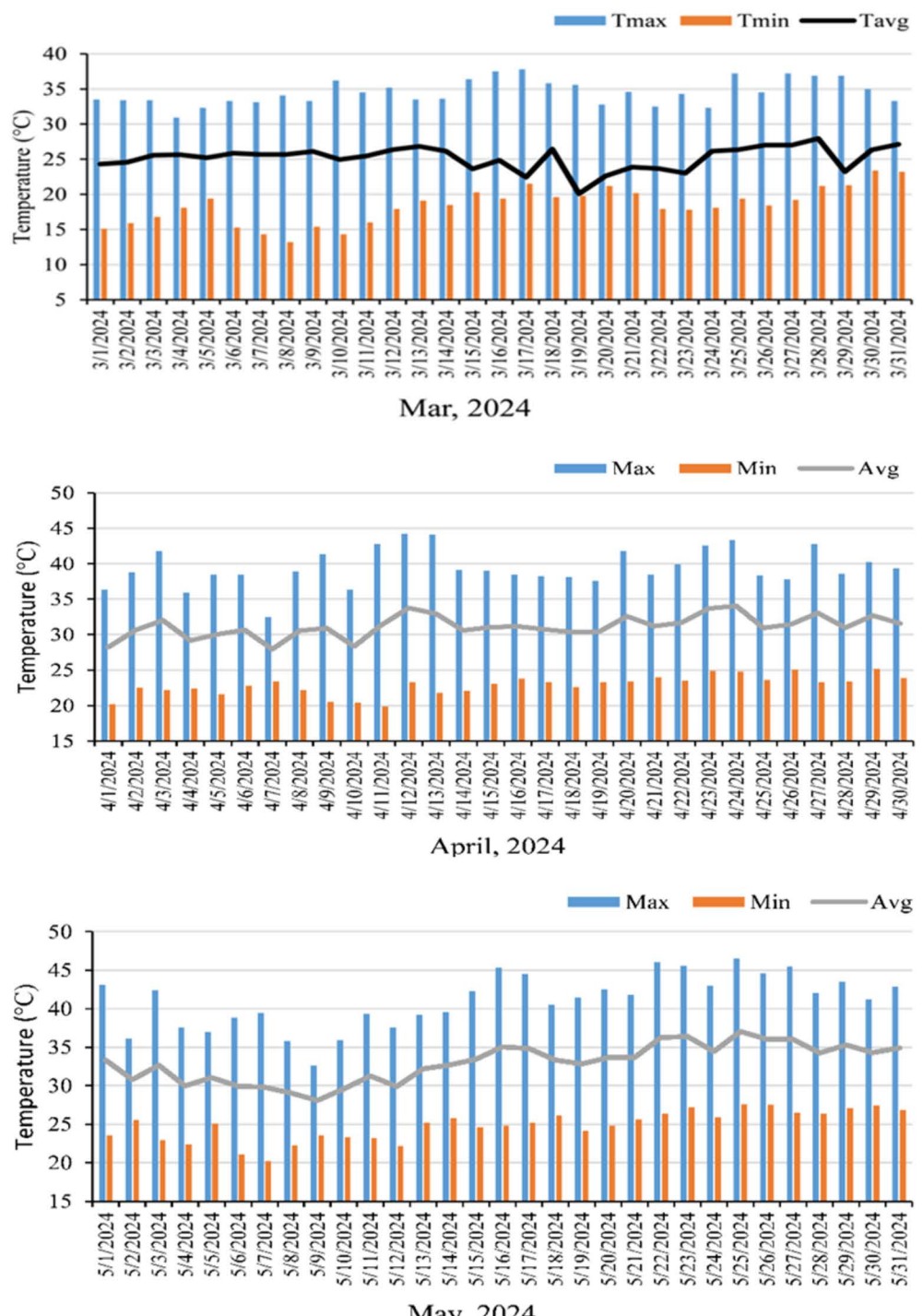

**Fig 4. maximum, minimum and daily average air temperature (°C) of the experimental field in the month of March, April and May, 2024.**

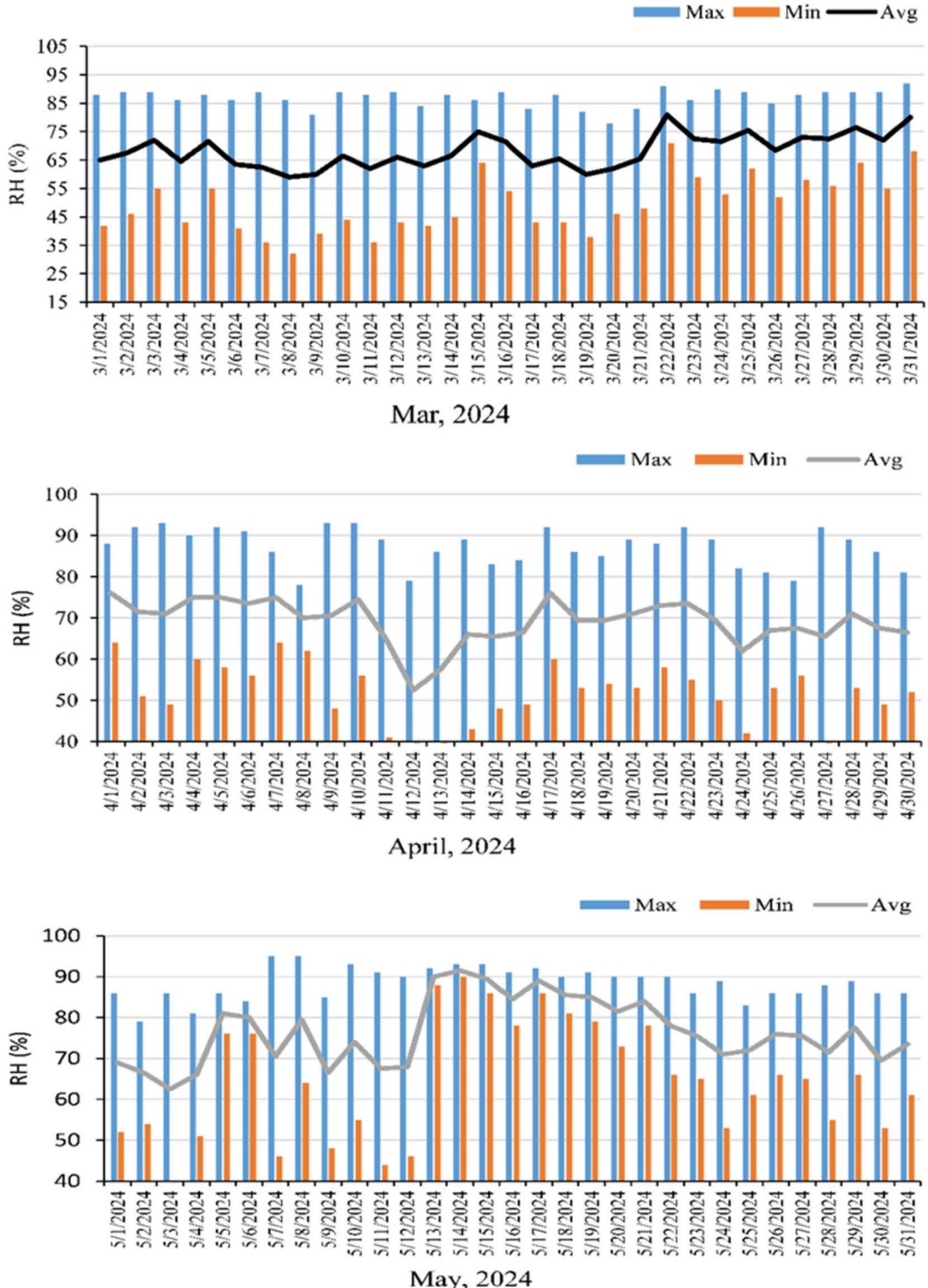

**Fig 5. maximum, minimum and daily average air relative humidity (RH %) of the experimental field in the month of March, April and May, 2024.**

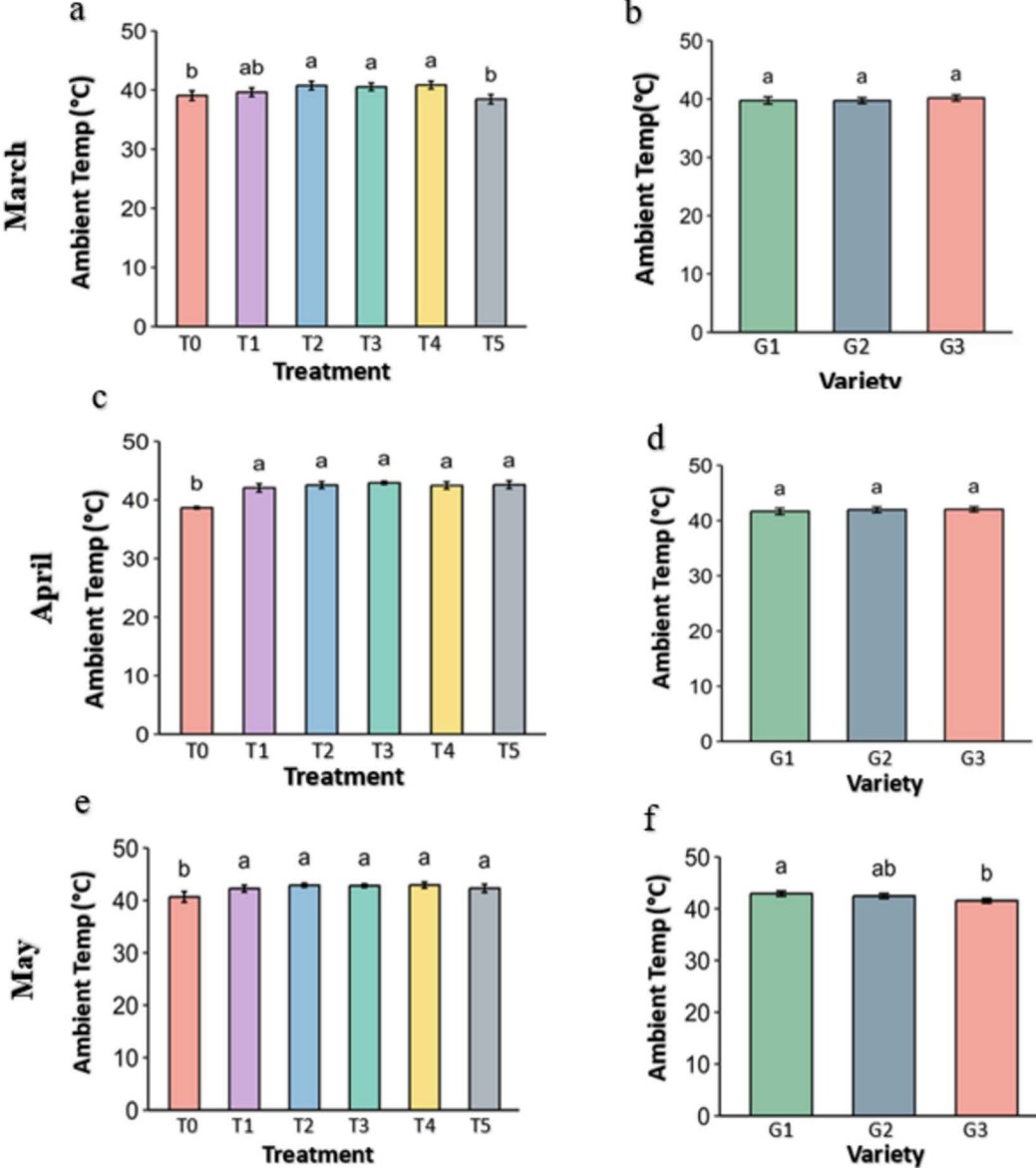

**Fig 6. ambient temperature (°C) on the leaf surface of strawberry plants under different treatment and varieties.** Treatment and genotypes having different letter varies significantly at p ≤ 0.05.

a, c, and e show the effects of different genotypes under various treatments and panel b, d, and f show the overall effects of treatments averaged across genotypes (Fig 6–18.). The data were obtained from the fluorometer in all the plants of three genotypes and analyzed at p ≤ 0.05. The result shows that leaf of plant from Control (T0) and Abscisic acid (T5) treatment had slightly lower temperature while leaf under other treatments has slightly higher temperature in March.

In both April and May, the surface temperature was slightly higher in the treatments than Control (38.7 °C in April and 40.7 °C in May). The leaf surface temperature of the genotypes didn't vary among them in March and April but it was slightly lower in the G3 genotype.

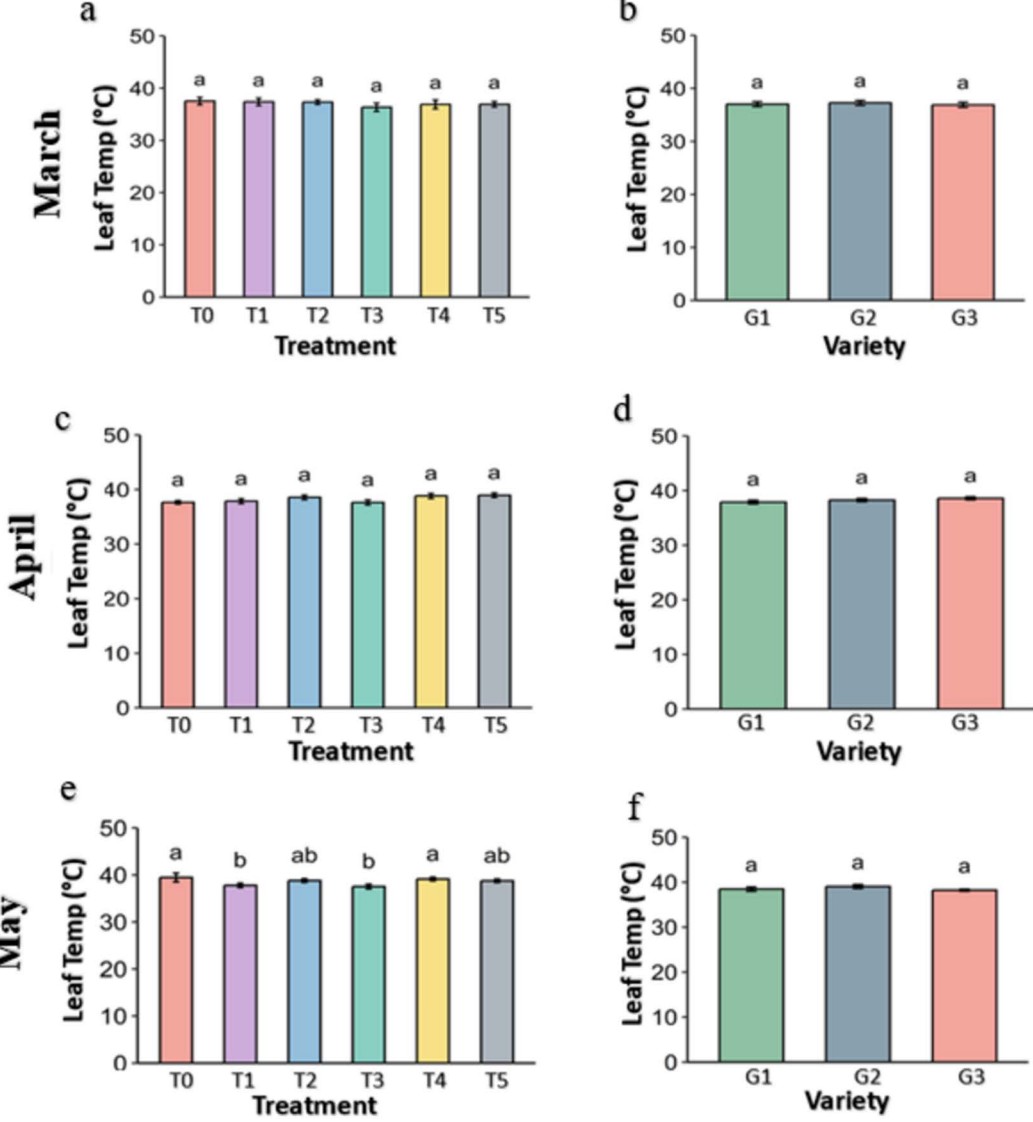

**Fig 7. leaf internal temperature °C of strawberry plants under different treatments and genotypes.** Treatment and genotypes having different letter varies significantly at p ≤ 0.05.

The leaf internal temperature under the treatments were reduced in all the treatments than surface ambient temperature but did not vary in March and April in comparison to the Control (37.5 and 37.7 °C respectively), later on, which reduced significantly in April than 39.4 °C in control (Fig 7). The leaf internal temperature was lowest in Kaolin (T3) (37.5 °C) followed by Molasses (T1) (37.9 °C). The genotypes showed no significant differences among them in any month.

The plant response under the heat stress govern by the treatments and genotypes are best depicted by the temperature differential (difference between surface temperature and leaf internal temperature). All the treatments reduced the leaf internal temperature significantly in compared to the control plants in all three leave of heat stress in three month (Fig 8). The Kaolin (T3) treatment reduced internal temperature by −4.2 °C in March, −5.3 °C in April and −5.3 °C in May in compare to the −1.5 °C, −1 °C and −1.2 °C reduction in the control. The Kaolin (T3) treatments were followed by Molasses

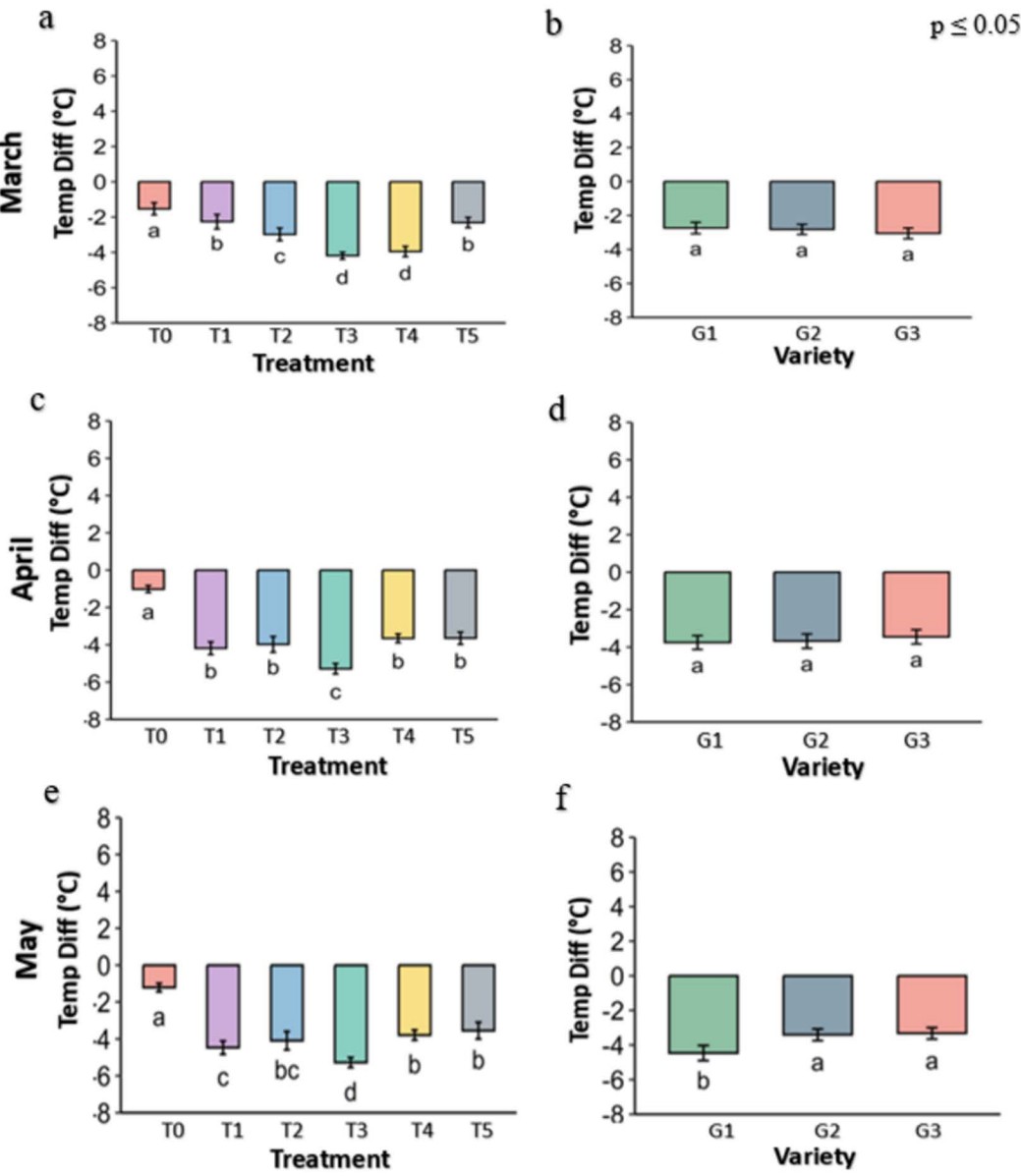

**Fig 8. leaf temperature differential °C of strawberry plants under different treatments and genotypes.** Treatment and genotypes having different letter varies significantly at p ≤ 0.05.

(T1) and CaCl2 (T4) for the internal temperature reduction ability. The temperature reduction capability of the genotypes was almost similar and didn't vary significantly in March and April, but in May in G1 genotypes.

### 3.3 Leaf attributes of the strawberry plant

Two important leaf attributes like leaf thickness (Fig 9) and leaf greenness (SPAD) were measured with the help of the fluorometer of all the plants of all genotypes and significance was analyzed at p ≤ 0.05. In March, the leaf thickness (μm) was not significant among the treatments but in April and May it was significantly higher in the treatments than Control. In Control (T0), the leaf thickness was 0.23 μm in March and April, which reduced to 0.17 μm in May. On the other hand,

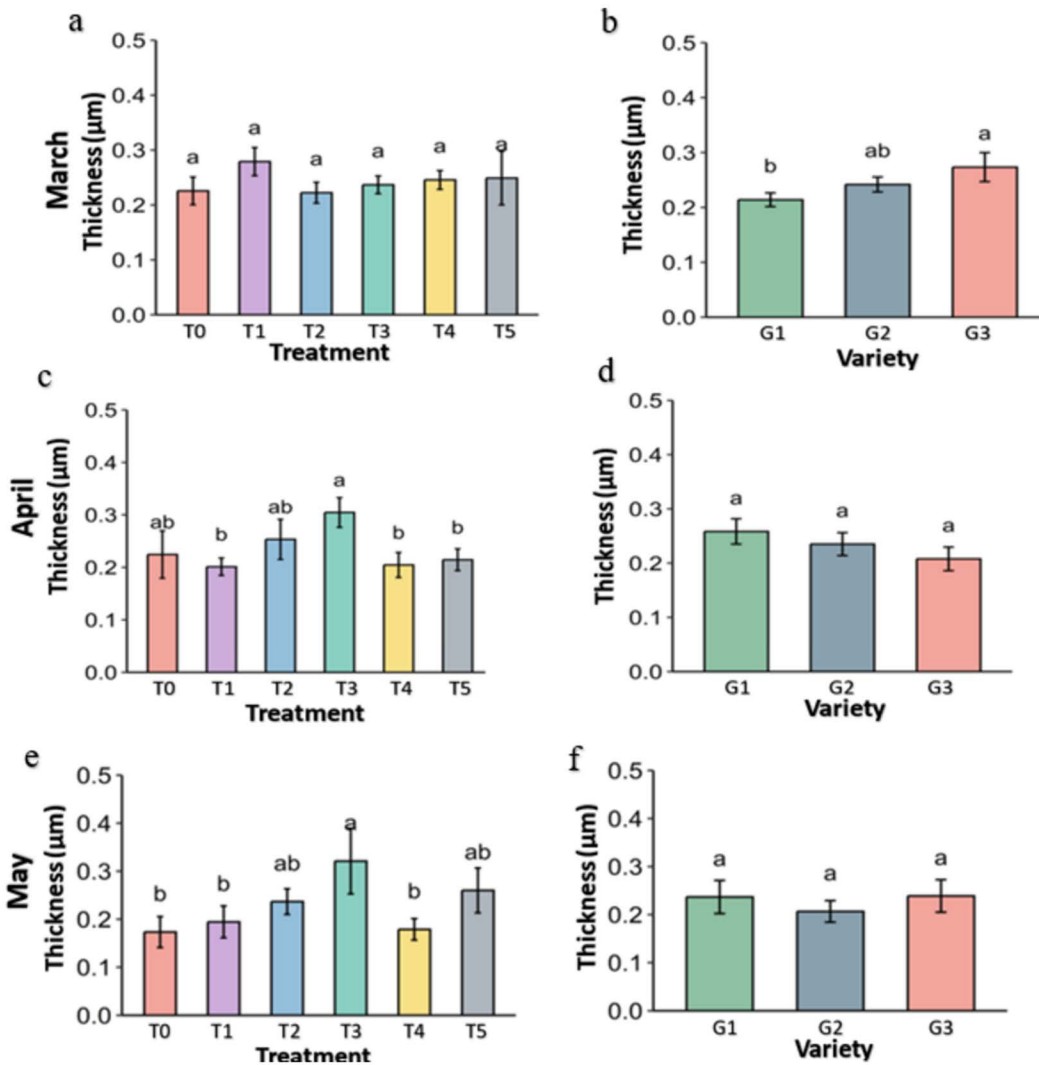

**Fig 9. leaf thickness (µm) of strawberry plants under different treatments and genotypes.** Treatment and genotypes having different letter varies significantly at p ≤ 0.05.

leaf thickness in Kaolin (T3) was 0.25 µm in March that increases to 0.30 µm in April and 0.32 µm in the May. Among the genotypes, leaf thickness of G3 was significantly thicker in the March (0.27 µm) but reduced to 0.21 µm in April and 0.24 µm in May while it was 0.21 µm, 0.24 µm, and 0.23 µm in the G1.

The greenness of the leaves is an indicator of the green pigments content, that is the main apparatus of the photosynthesis system. The SPAD (leaf greenness) of the leaves under the treatments were higher in March but not significant than control but became significantly higher in April and May (Fig 10). Actually, SPAD (leaf greenness) didn't increases over the period in the treatments but preserved with slight decrement, while control loss the greenness of the leaves over the months drastically (42. 1 in March, 34.1 in April and 36.4 in May). Melatonin (T2), CaCl2 (T4) and Abscisic acid (T5) perform better to preserve the greenness over the time. In genotypes, leaf greenness was significantly higher in G1 in March (44.6) and April (43.7) than others but become almost level by May (41.9 in G1, 42.8 in G2 and 42 in G3).

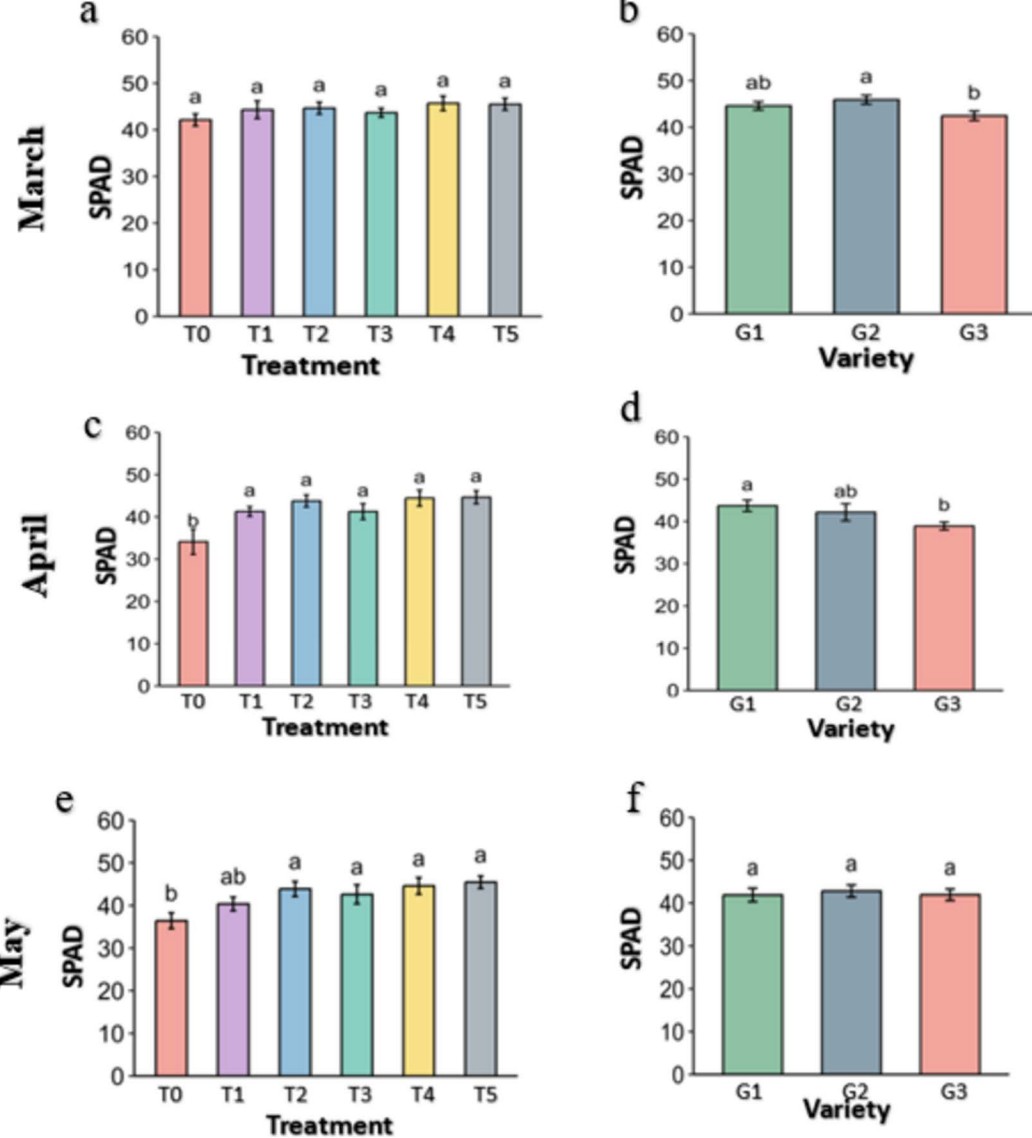

**Fig 10. leaf greenness (SPAD) of strawberry plants under different treatments and genotypes.** Treatment and genotypes having different letter varies significantly at p ≤ 0.05.

### 3.4 Leaf fluorescence of the strawberry plant

Maximum quantum efficiency of Photosystem II denoted as Fv/Fm is a widely used indicator to observe the stress level in the plants. It is a proportion value which can be maximum of 0.83 and the higher the value, the health condition of the plant is better and vice versa. In March, plants under Abscisic acid (T5) show significantly better maximum quantum efficiency than other treatments (Fig 11). In April and May, the Maximum quantum efficiency was reduced in control drastically with some other treatments but Kaolin (T3) has highest Maximum quantum efficiency followed by Molasses (T1) and Abscisic acid (T5). In the genotype, Maximum quantum efficiency didn't vary in March but reduced over the time in all genotypes and most reduction was seen in G2 genotype while least variation was observed in G3 genotype.

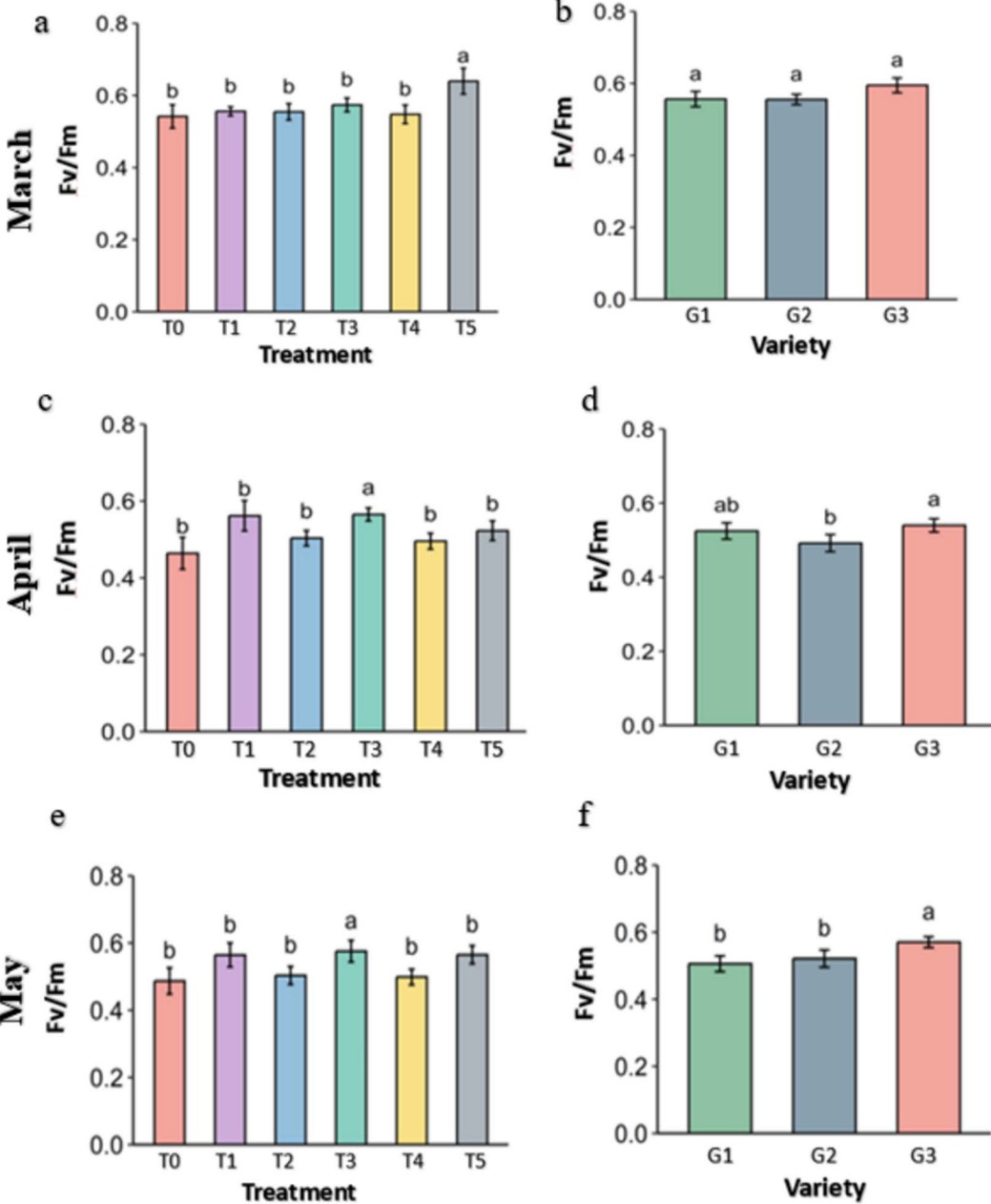

**Fig 11.  maximum quantum efficiency (Fv/Fm) of strawberry plants under different treatments and genotypes.** Treatment and genotypes having different letter varies significantly at p ≤ 0.05.

Linear electron flow (LEF) in the photosystem II is an indicator of photosynthetic activity in the plant leaves. This parameter is a product of photosynthetically active radiation (PAR) received by the leaves, the quantum yield produced by the leaves from the harvested light and a coefficient given by the protocol. The LEF was significantly higher in Melatonin (T2), followed by Molasses (T1), CaCl2 (T4) and Abscisic acid (T5) in the heat stress of March (Fig 12). By April and May, LEF in the control drastically reduces and was significantly higher in the treatments but highest in Abscisic acid (T5) followed

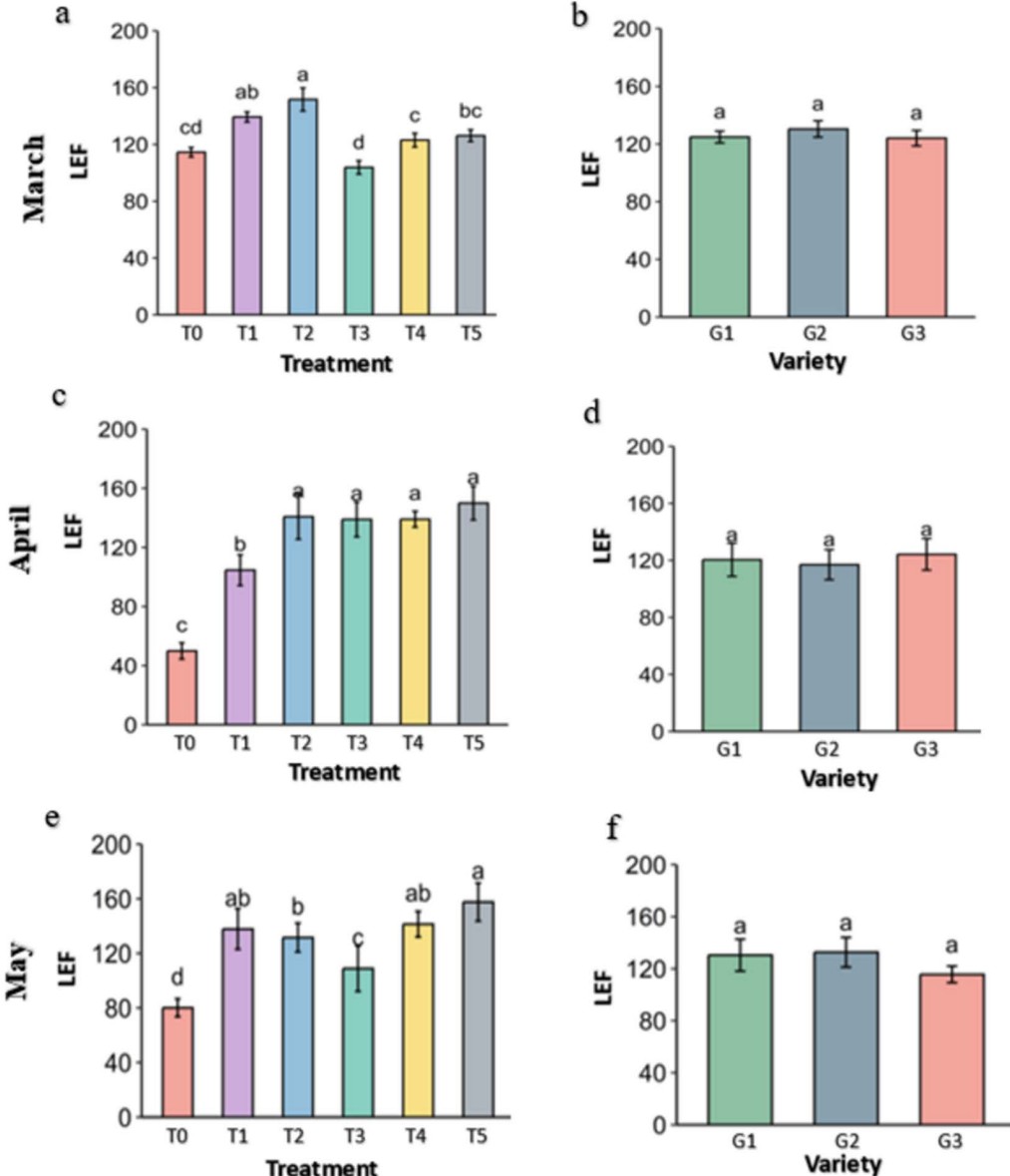

**Fig 12. linear electron flow (LEF) of strawberry plants under different treatments and genotypes.** Treatment and genotypes having different letter varies significantly at p ≤ 0.05.

by CaCl2 (T4), Melatonin (T2) and Molasses (T1). The treatments didn't increase the LEF but preserved from the reduction like control. The genotypes didn't vary significantly among them in terms of LEF but G1 has slightly higher LEF than others.

Quantum yield (φII) is the product of photosystem II (PSII) takes place in the leaves during photosynthesis, is an indicator of plant health under the condition. The quantum yield (φII) was significantly higher in the Abscisic acid (T5) treatments than other in March, but under the heat stress on April and May Molasses (T1), Kaolin (T3) and Abscisic acid (T5) preserved the Quantum yield (φII) significantly while drastic reduction was observed in the control (Fig 13). In genotypes, G3

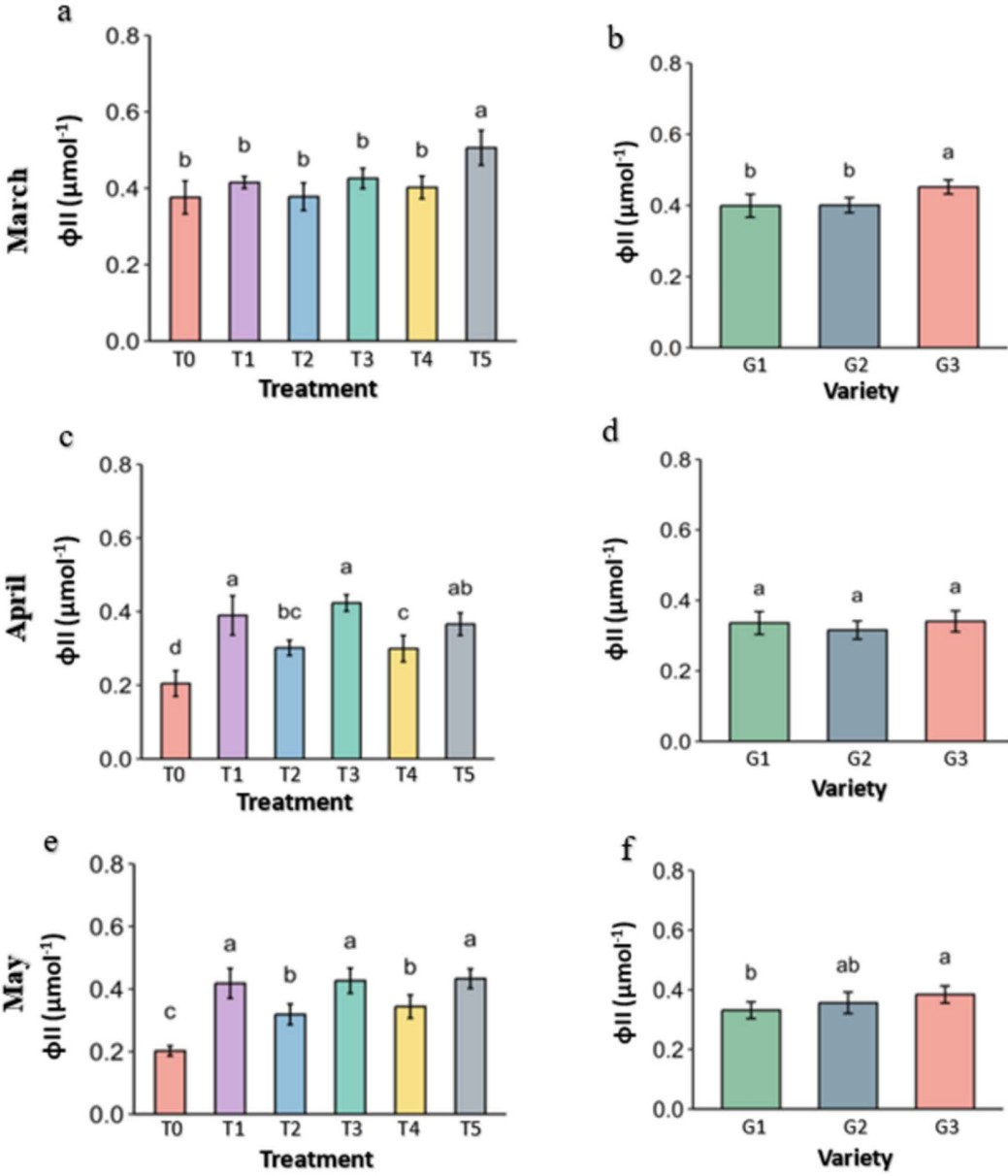

**Fig 13. quantum yield (φII) of PSII of strawberry plants under different treatments and genotypes.** Treatment and genotypes having different letter varies significantly at p ≤ 0.05.

has significantly higher Quantum yield (φII) in March but get reduced by April and May, becomes similar to the G1, which didn't loss much yield over the stress period.

The amount of Non-regulatory Energy loss through dissipation in the heat form from the PSII is denoted as φNO. The excess energy dissipation from the system was consistent from month to month and was not varied significantly $p \leq 0.05$ among the treatment, not the genotypes (Fig 14).

The harvested light which plant can't utilize due destruction in apparatus of PSII is released from the leaves by regulated Non photochemical quenching (φNPQ). The amount of Non photochemical quenching is directly associated with the

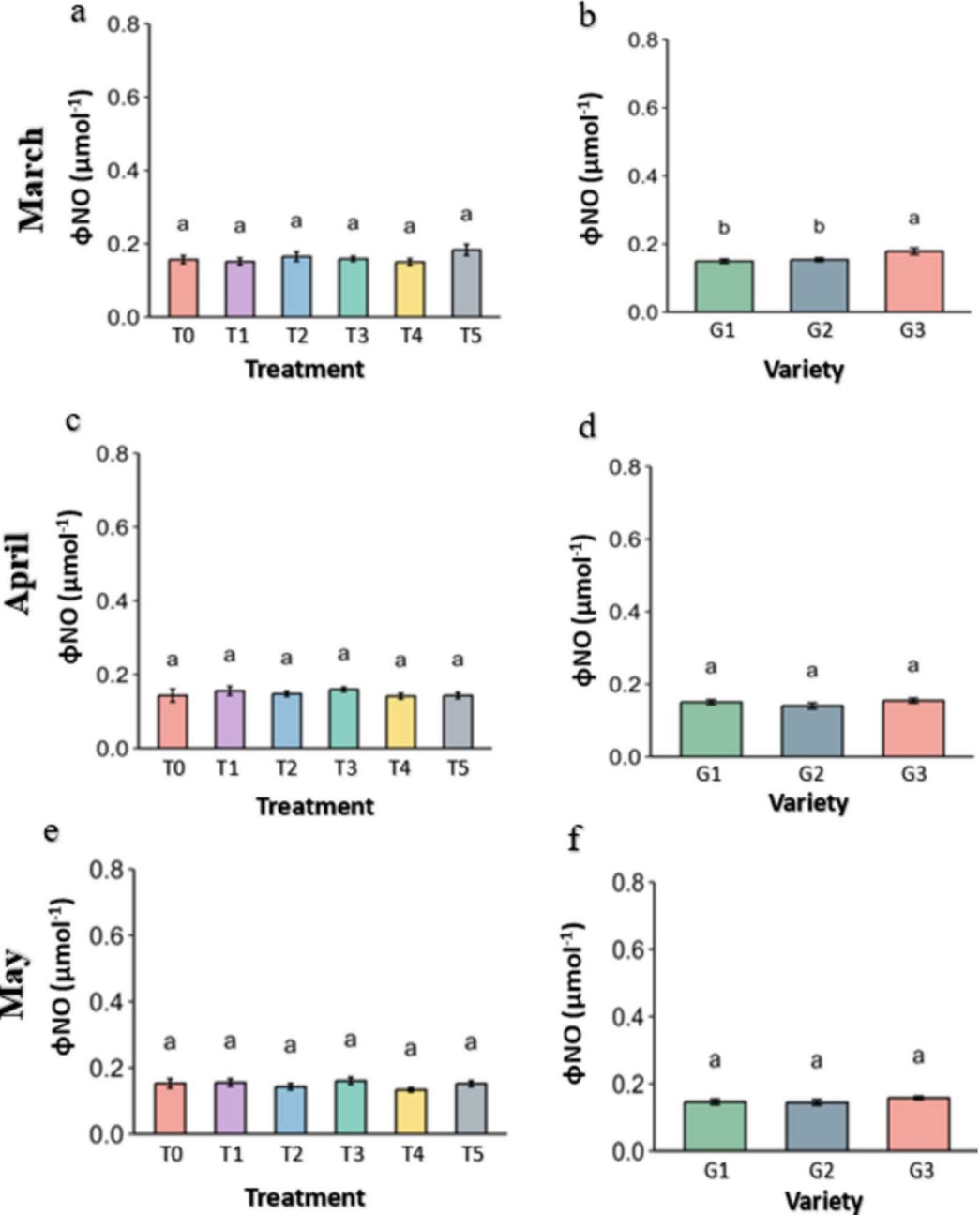

**Fig 14. non-regulatory energy dissipation (φNO) of PSII of strawberry plants under different treatments and genotypes.** Treatment and genotypes having different letter varies significantly at $p \leq 0.05$.

stress level of the plant. It was significantly lower in the Abscisic acid (T5) in the month of March in comparison to the Control (Fig 15). By the April and May, it was stable and didn't increase in Molasses (T1), Kaolin (T3) and Abscisic acid (T5) treatments while increased in other treatments with maximum in control (0.47 in March, 0.65 in April and 0.65 in May). In the genotypes, it was significantly lower in G3 in March but become level with the G1 through increment while G1 was stable. By May, φNPQ decreases slightly in G3 genotypes.

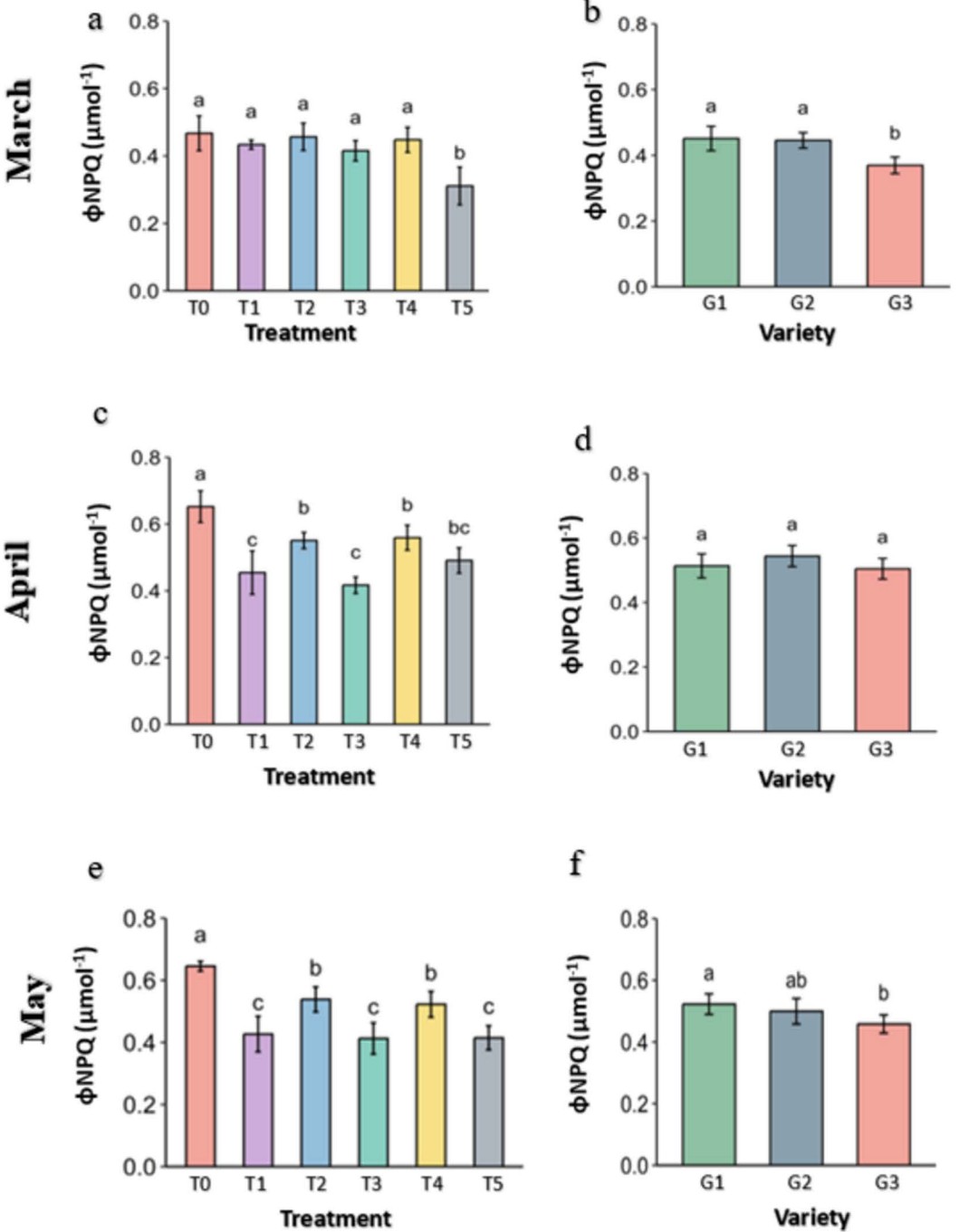

**Fig 15. regulated non photochemical quenching (φNPQ) of PSII of strawberry plants under different treatments and genotypes.** Treatment and genotypes having different letter varies significant at $p \leq 0.05$.

Photosystem II Redox State (qL) describes activity of the plant cell for the proper functioning. The higher the qL is the PSII apparatus are functioning better for carrying out the photosynthetic activity. The qL was significantly lower in control followed by Melatonin (T2) in March while it was better in Molasses (T1), Kaolin (T3), CaCl2 (T4) and Abscisic acid (T5) treatments (Fig 16). The qL was further reduced in the control in April and May subsequently while the value was

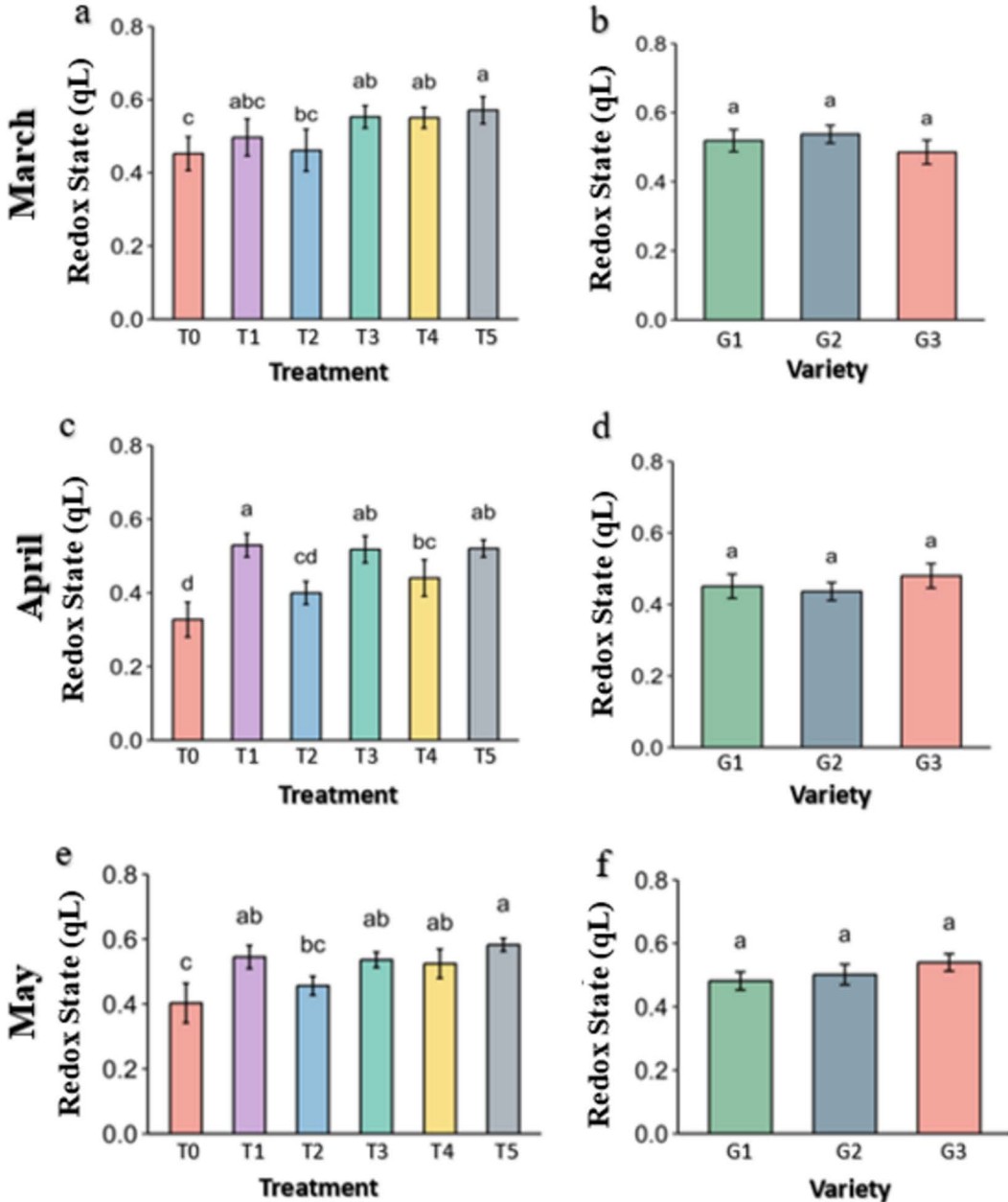

**Fig 16. redox state (qL) of PSII of strawberry plants under different treatments and genotypes.** Treatment and genotypes having different letter varies significantly at p ≤ 0.05.

preserved in Molasses (T1), Kaolin (T3), CaCl2 (T4) and Abscisic acid (T5) were significantly higher than the Control. In genotypes, though difference was not significant among them, the G1 genotype has slightly better qL in March.

### 3.5 Proton activity of the strawberry plant

Steady-State Proton Flux (vH) is the indicator of $H^+$ transportation in the photosynthetic system during the photosynthesis under light. It was almost similar among the treatments in comparison to control in March (Fig 17). But vH decreases drastically in the control in alleviate heat stress in April and May, while preserved in all the treatments. However, Melatonin (T2)

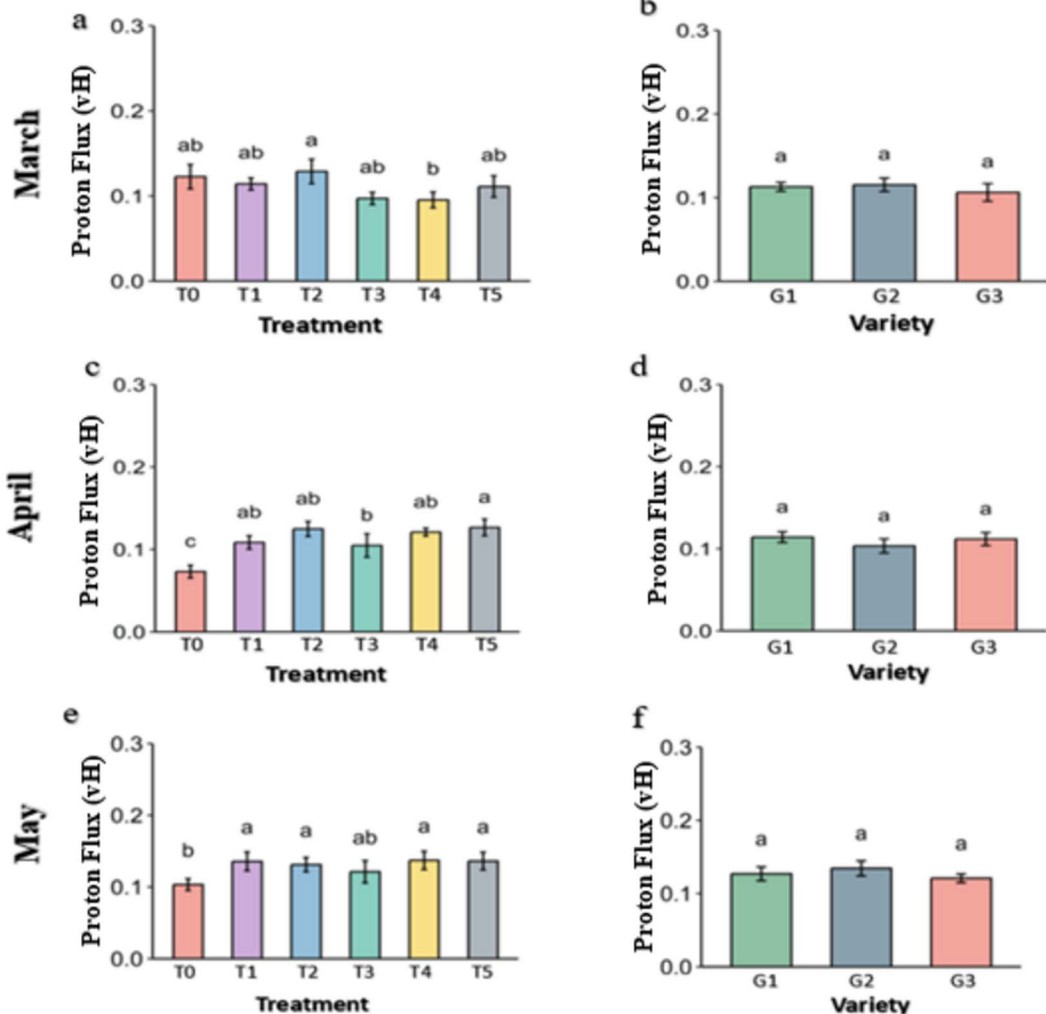

**Fig 17. steady-state proton flux (vH) of PSII of strawberry plants under different treatments and genotypes.** Treatment and genotypes having different letter varies significantly at p ≤ 0.05.

and Abscisic acid (T5) seems to have slight better proton flux than others. In the genotypes, vH was not significant among the treatments but G1 has slight better vH.

Proton conductivity (gH) was significantly higher in the Abscisic acid (T5) treatment than others in March ([Fig 18]). By the April and May, gH drastically decreased in the control and preserved in the treatments with higher in Molasses (T1), Kaolin (T3) and Abscisic acid (T5). In genotypes, conductivity was higher in G3 than others in March and April but it increases in G1 and decreases in G3 by May and become similar.

### 3.6 Pearson correlation between the variables

Photosynthetically active radiation (PAR) is one of the mandatory elements for the Photosynthesis to be carried out that ignite the process. Pearson's correlation suggest that PAR positively significantly increases the leaf surface ambient temperature, leaf internal temperature, linear electron flow in leaves, steady-state proton flux as well as regulatory-non photochemical quenching. Fv/Fm showed a strong positive correlation with SPAD (r = 0.83, p <

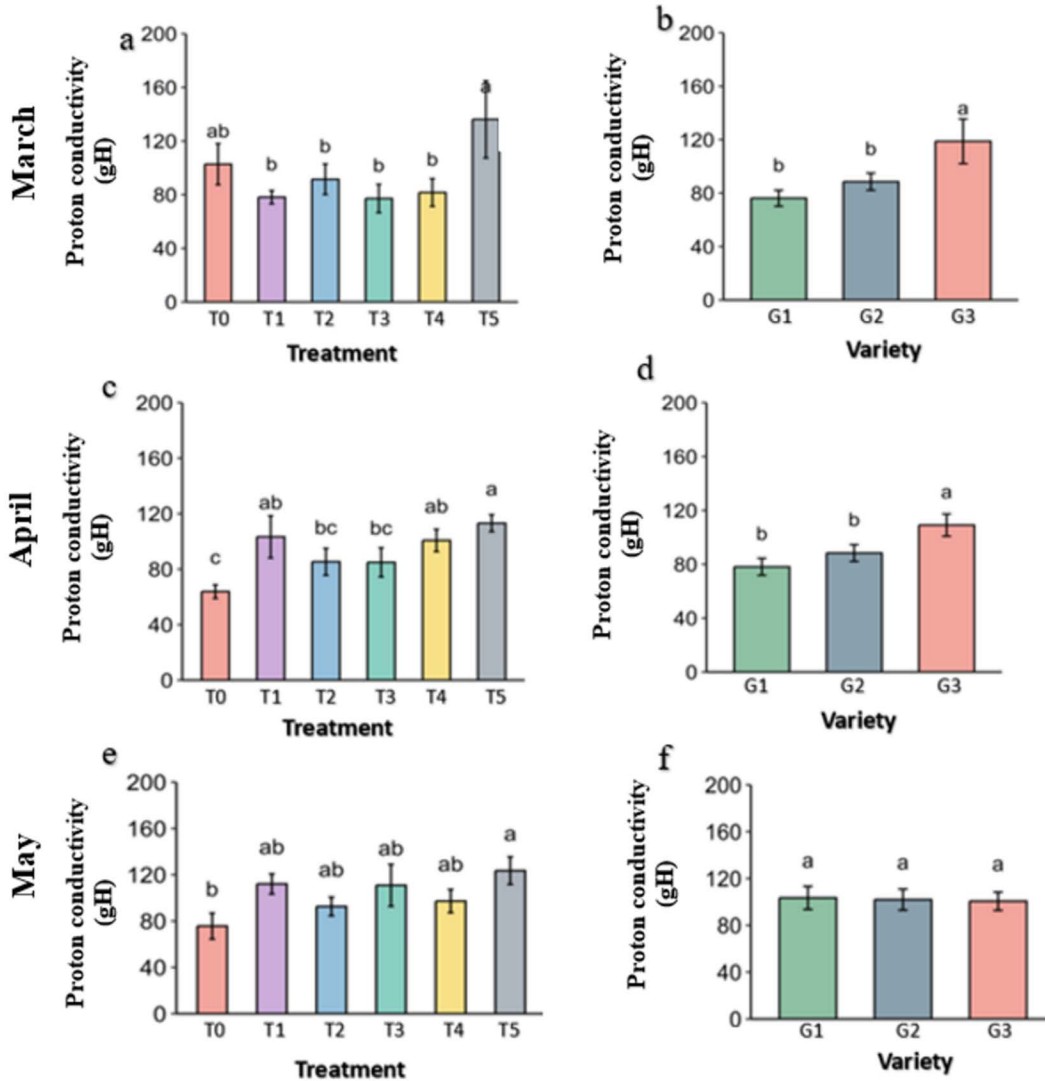

**Fig 18. proton conductivity (gH) of PSII of strawberry plants under different treatment and genotypes.** Treatment and genotypes having different letter varies significantly at p ≤ 0.05.

0.001) andPhi2 (r = 0.71, p < 0.001), indicating greater chlorophyll content and photochemical efficiency supported more effective PS II performance. Similarly, LEF and vH showed positively correlations with Phi2 (r = 0.71, p < 0.001) and vH (r = 0.60, p < 0.001), indicating that electron and water transport were more efficient under the optimum environment. In contrast, PhiNPQ had a negative correlation with Phi2 (r = −0.62, p < 0.001) and Fv/Fm (r = −0.58, p < 0.001), indicating increased energy dissipation under heat stress. Additionally, ambient and leaf temperatures showed negative associations with photosynthesis variables, especially with Phi2 and Fv/Fm, reflecting heat induced physiological limitations (Fig 19).

Treatment wise phenotypic variation regarding growth and development of different strawberry genotypes were given below (Fig 20)

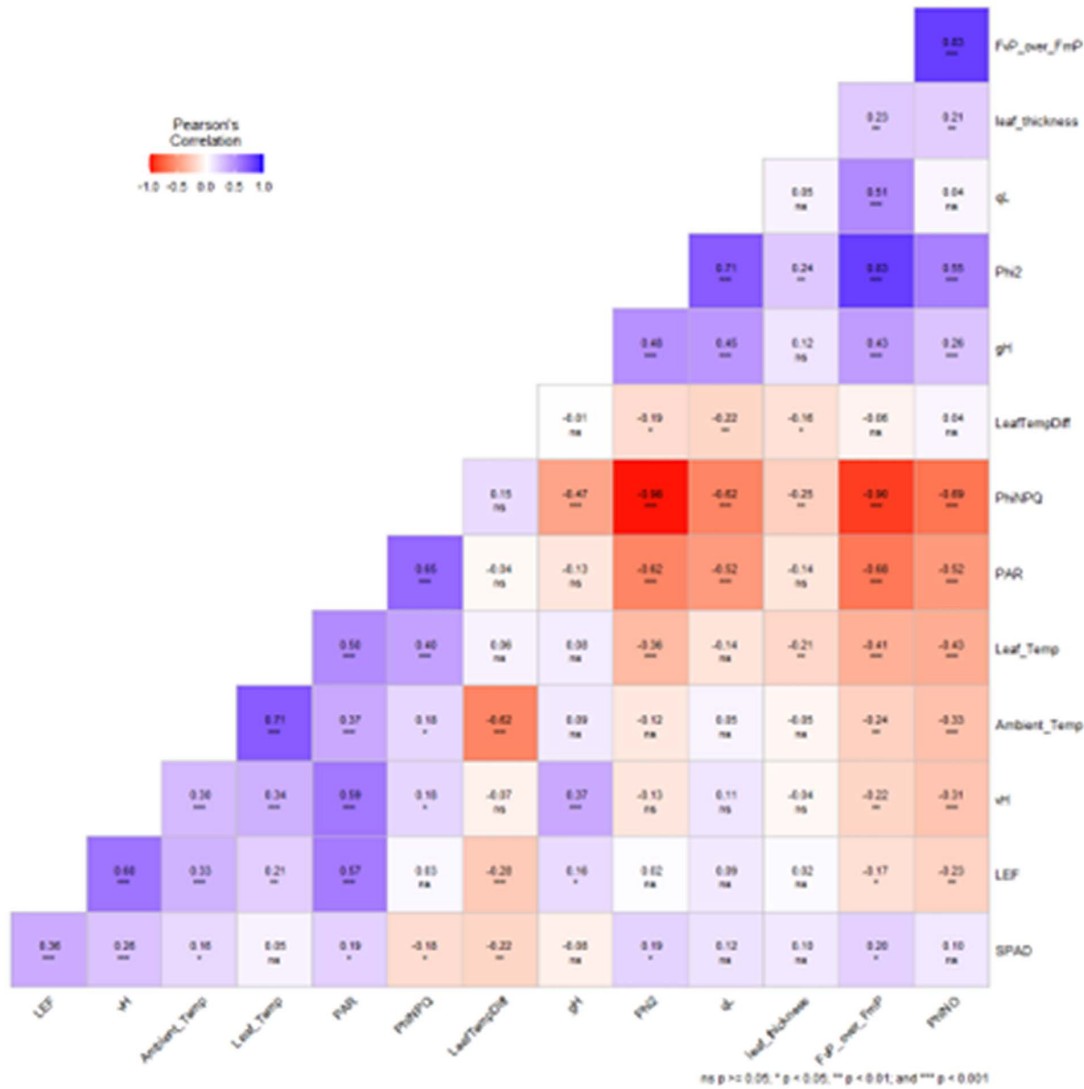

**Fig 19. Pearson's correlation between the parameters of heat stress tolerance in strawberry plants under different treatments and genotypes.** A positive sign relates to positive correlation and negative sign relates to negative relation. Here, "***" is significant at 0.001, "**" is significant at 0.01 and "*" is significant at 0.05 p value. Parameters are, "LEF" Linear electron flow, "vH" = S eady state proton flux, "Ambient_Temp" = L af surface ambient temperature, "Leaf_Temp" = L af internal temperature, "PAR" = Photosynthetically active radiation, "PhiNPQ" = R gulated non-photochemical quenching, "LeafTempDiff" = Temperature differential between surface and internal, "gH" = Proton conductivity, "Phi2" = Quantum yield of PSII, "qL" = P II redox state, "Leaf_thickness" = Thickness of the leaf, "FvP_over_FmP" = Maximum quantum efficiency of PSII, and "PhiNO" = Non-regulatory energy dissipate.

## Discussion

March is the start of the summer in Bangladesh when both day and night temperatures start to rise and exceed 35 ℃ at day and 20 ℃ at night. The temperature intensified in April which fluctuated around 40 ℃ but remain higher than 35 ℃ due to sporadic rain. At May, heat waves can keep the air temperature above 40 ℃ for a few days to a prolonged period. At the onset of summer at march it was around 49.5% daily average but increased to 69% in April and become highly humid of 76% daily average at May. Humid conditions do not dry out wet surfaces thus leaf temperature might rise. Kadir and colleagues [29] demonstrated that leaf temperature increased under 40/35'C, reaching an average of 39.3 ºC within a

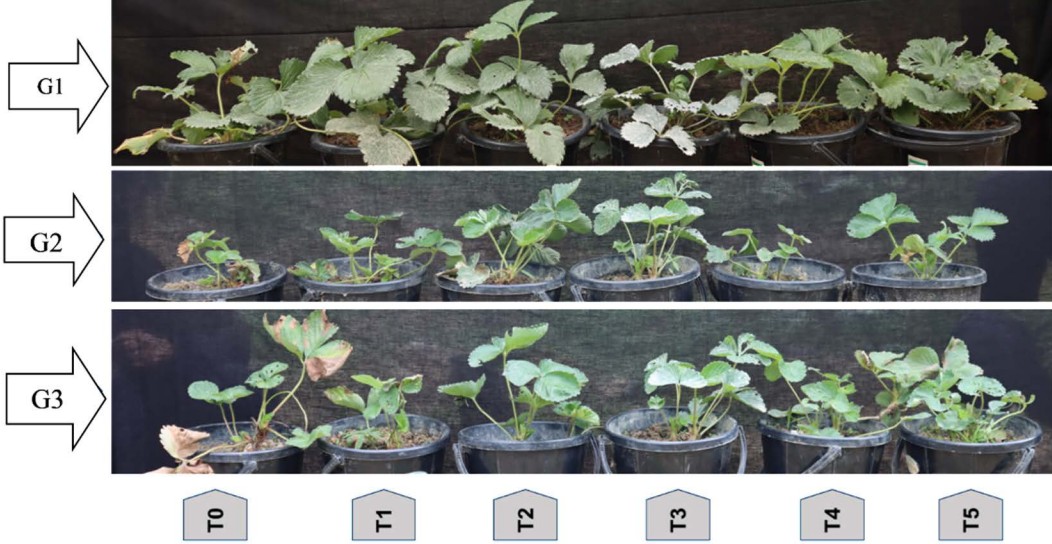

**Fig 20. The difference between the genotypes and the effect of the treatments on the plants.**

week in two cultivars (Chandler and Sweet Charlie), indicating heat stress impact on the cultivars. Our study supports this, showing the ambient leaf surface temperature was high in all the plants but it was comparatively slightly higher in the treatments than Control (T0) and Abscisic acid (T5). This one might seem conflicting since kaolin and molasses are reflective or coating materials that affect surface-level infrared measurements but not always internal leaf temperature. To the contrary, ABA and water sprays left no residues, but since no reflectance was altered, they have slightly lower surface temperature. The shiny surface of the G3 might be a reason to slightly lower surface temperature. The application of Kaolin (T3) and Molasses (T1) leave a coating trace on the leaf surface, which might help to reduce the leaf internal temperature. Plants naturally keep its leaves' internal parts cooler than the surface temperature for proper functioning. These capabilities of cooling increase upon aid with the applied exogenous substances. However, the higher differential in Kaolin (T3), and molasses (T1) reduction might be due to thick coating on leaves and in $CaCl_2$ (T4) might be due to stomatal regulation. The tolerant genotype G1 is better at cooling the internals and maintaining the differential, likely due to intrinsic thermoregulation mechanisms. The reduction in the thickness in the leaves of the control plant might be associated with impaired photosynthetic activity or the heat stress-reducing mechanism of the plants. The increment in the thickness in plants under Kaolin (T3) application might be due to the improvement of the health of leaves or the deposition of a layer on the surface of the leaves. The leaves of G3 were genetically thicker but became thinner might be due to impaired photosynthetic activity. The SPAD (leaf greenness) level was almost similar in the treatments at the onset of summer but it reduces with the heat stress in the upcoming months. The lessening of SPAD (leaf greenness) denotes destruction of pigments and photosynthetic apparatus. Kadir and colleagues [29] similarly reported a 21% decrease in SPAD in Sweet Charlie after one week of heat exposure and 13% in Chandler after two weeks. In our study, preservation of SPAD (leaf greenness) in Melatonin (T2), $CaCl_2$ (T4) and Abscisic acid (T5) might be associated with the internal cell regulation process by these chemical mitigating substances. The greenness was higher in the tolerant genotype (G1) signifies the importance of better greenness for heat tolerance. The maximum quantum efficiency of Photosystem II indicates the stress condition of the plants [30]. Under the stress, Fv/Fm reduced drastically in the control but kaolin (T3), Molasses (T1) and Abscisic acid (T5) is capable of preventing the reduction, might be associated with preservation of greenness in the leaves. It was lower in susceptive G2 genotype suggest G2 is most pruned to heat stress as neither it does have own tolerant property nor response to mitigating substance properly. Rashid and colleagues [31], found that

extended heat stress led to a progressive decline in chlorophyll fluorescence by 5.1% after 7 days and up to 43.7% after 28 days compared to ambient conditions. These highlight the cumulative damage caused by prolonged high temperature. Linear electron flow (LEF) in is the product of photosynthetically active radiation (PAR) received by the leaves, the quantum yield produced by the leaves from the harvested light and a coefficient given by the protocol [27]. The LEF reduces in control might be related to PS apparatus damage and preserving in Melatonin (T2), Molasses (T1), CaCl$_2$ (T4) and Abscisic acid (T5) might be due to internal regulation of the cells. The tolerant genotype has better LEF as it can utilize PAR more efficiently. Similar to Zhang and Sharkey [32], who showed reduced LEF in pgr5 mutants under high temperature due to impaired cycle electron flow regulation. Quantum yield (φII) of photosystem II (PSII), another critical parameter, is an indicator of plant health under the stress [30], which declined sharply with the increment of heat in control due to the damage of PSII system. Treatment like Molasses (T1), Kaolin (T3) and Abscisic acid (T5) reduces leaf internal temperature, or regulates the cell to keep up the quantum yield under the stress. Exogenous spray might increase tolerance in G3 genotypes but does not sustain under high heat but tolerant G1 genotypes can keep the yield due to own mechanism. High temperatures lead to photoinhibition of PS II, and subsequent decreased rates of carbon assimilation, resulting in a reduction in photosynthesis, as found in Maize [33]. Lower net photosynthesis and less chlorophyll fluorescence, as Fv/Fm, after 7 days of heat stress further verify damage to PS II and established stress in the leaves [34]. Non photochemical quenching (φNPQ) is a good indicator assessing impact of stress [30,27,35]. NPQ increased under heat stress, especially when Rubisco deactivation limited electron transport. This was attributed to the need for dissipating excess light energy as heat [36]. Experiment on Arabidopsis leaves showed that NPQ increased more at 44 than at 40, and was significantly higher after 30 min of heat stress compared to only 4 min, reflecting rapid stress response [37]. When high heat damage the leaf organelles, plant cannot process all the harvested light to produce the food. Thus, leaves expelled excess light or energy through non-photochemical quenching. The higher the damage, the higher the φNPQ from the leaves. In our findings, a lower φNPQ in Abscisic acid (T5) implies better utilization of light in photochemical pathway. Along with Abscisic acid (T5), Kaolin (T3) and Molasses (T1) application increases better utilization of light in photochemical pathway. The stability of φNPQ in G1 further highlights its tolerance capability with the heat and lower in G3 at early summer and increment in late summer denotes exogenous substance support the susceptible plant up to a level, not more. A proper functioning machinery in the plant cell result in high photosystem II Redox State (qL) [38]. The lower qL value in control implies less activity and uses of photosynthetic machinery in the leaves. However, Molasses (T1), Kaolin (T3), T4 (CaCl$_2$) and Abscisic acid (T5) support the PS machinery and proper activity in the pathway results in better qL. The tolerant G1 genotypes has better qL for better activity at the pathway. The parameter qL, a measure of the fraction of open PS II centers, showed a significant decrease under heat stress, so plastoquinone became more oxidized [37], further confirming the impact of heat on photosynthetic efficiency. The linear electron flow and redox state of the cell PS II is related to the Steady-State Proton Flux (vH), indicating mobilization of H$^+$ in the photochemical pathway [39]. Likewise, Melatonin (T2) and Abscisic acid (T5) is better at regulating cell internal process thus results in better vH as well as in tolerant G1 genotypes. The same impact of Abscisic acid increases the proton conductivity (gH) in the leaves. From the correlation analysis, it is apparent how heat stress can cause physiological changes in plants, particularly to the photosynthetic apparatus. The strong negative correlations between both ambient and leaf temperatures with Phi2 and Fv/Fm indicate that under elevated thermal conditions, photosystem II efficiency is negatively impacted and that photochemistry would be impaired by reduced photochemical energy conversion efficiency. This result aligns with previous research that high temperature leads to structural changes in the thylakoid membrane and impose damage to the protein of photosystem II, leading to a decrease in quantum yield [37]. Given the strong negative correlation between PhiNPQ and both Phi2 and Fv/Fm, the response is indicative of an enforced shift away from photochemistry during heat stress, to non-photochemical quenching, which is a protective mechanism that dissipates excess excitation energy as heat. This response is important for avoiding photoinhibition but also necessarily reduces the energy available for photochemistry. Moreover, the positive association between SPAD and Fv/Fm supports the contribution for chlorophyll content to sustaining PS II function under heat stress.

The strong association of Phi2 with both LEF and vH illustrates the way in which coordinated water transport and electron flow can ensure efficient photosynthesis under heat stress. Efficient water transport may allow the leaf to maintain a cooler temperature via transpiration and a capable electron transport chain ensures consistent energy flow while approaching thermally limited photochemistry. PAR had a positive correlation with LEF, but a negative correlation with PhiNPQ, indicating that light levels supported photochemistry until a limit, at which point energy could be dissipated as non-photochemical quenching as temperature increases from heat stress. The results show that plants that were exposed to heat stress for a prolonged period experiences phenomena that indicate both protective (NPQ) and compensatory (maintained chlorophyll, LEF and vH) mechanisms to avoid damage, but ultimately damage builds over time or with intense heat stress if the limits of the plant's protective and compensatory capabilities are compromised. These physiological interactions emphasize the importance of selecting genotypes which have stable photochemical responses and capable efficacy transport system for better heat tolerance. Finally, while photosynthetically active radiation (PAR) is essential for photosynthesis, but excess PAR under heat can cause photodamage, leading to increased $\varphi$NPQ and decreased Fv/Fm and $\varphi$II. Thus, a balanced PAR is demanding for the plants. Heat stress of up to 36 °C reduced the light radiation efficiency significantly [40]. Plants attributes like thickness and SPAD (leaf greenness) is very important for the plant to lessen impact of the heat stress. Leaf thickness can increase the temperature differential thus cooling the leaf internals, prevent chlorophyll damage and maintain proper photosynthetic activity. The organic substance treatment those develop surface coating like kaolin and Molasses is very good for the development of temperature differential. The chemical substances like melatonin, $CaCl_2$ and Abscisic acid is good at regulating internal cell processes to keep up the photosynthetic activity.

## Conclusions

Strawberry production in warmer climates such as Bangladesh, is increasingly being challenged by extreme summer temperatures and unpredictable heat waves. This study found that the heat tolerant genotype (G1) maintained more photosynthetic performance, leaf greenness and internal cooling efficiencies over the heat-susceptible genotypes. Additionally, the kaolin and molasses treatments were the only treatments that decreased internal leaf temperatures by forming protective coating on leaves, while melatonin, $CaCl_2$ and abscisic acid treatments had direct regulatory effects on internal physiological processes (stomatal regulation, oxidative process and radioprotection etc) and were able to maintain photosystem efficiency due to the identified heat stress, The combination of heat tolerant genotypes with targeted exogenous treatments, found in this study, is a realistic approach to Strawberry production during summer in tropical regions. However, the study had certain limitations. This was conducted under pot based, field like conditions rather than open field planting which likely had some influence on microclimatic effects and root zone interactions. Only three genotypes and five treatments were examined and did not evaluate long-term yield and fruit quality. Future work should aim at validating these findings on a wide variety of genotypes and under open field growing conditions. Moreover, assessing the interaction between exogenous treatments and soil health under stress, will be essential. Lastly, it may also offer deeper insight into mechanisms of heat tolerance by exploring the molecular and gene expression response to these treatments.

## Supporting information

**S1 File. Data.**
(ZIP)

## Author contributions

**Conceptualization:** Sadia Shabnam Swarna, Dr. Md. Masudur Rahman, Dr. Sharifunnessa Moonmoon.
**Data curation:** Sadia Shabnam Swarna, Dr. Md. Masudur Rahman, Dr. Sharifunnessa Moonmoon.

**Investigation:** Najra-Tan-Nayeem Salwa, Sadia Shabnam Swarna, Dr. Md. Masudur Rahman, Dr. Sharifunnessa Moonmoon.

**Methodology:** Najra-Tan-Nayeem Salwa, Sadia Shabnam Swarna, Dr. Sharifunnessa Moonmoon.

**Project administration:** Dr. Sharifunnessa Moonmoon.

**Resources:** Dr. Md. Masudur Rahman, Dr. Sharifunnessa Moonmoon.

**Software:** Sadia Shabnam Swarna, Dr. Sharifunnessa Moonmoon.

**Supervision:** Dr. Sharifunnessa Moonmoon.

**Validation:** Najra-Tan-Nayeem Salwa, Dr. Md. Masudur Rahman.

**Writing – original draft:** Najra-Tan-Nayeem Salwa, Sadia Shabnam Swarna, Dr. Md. Masudur Rahman, Dr. Sharifunnessa Moonmoon.

**Writing – review & editing:** Sadia Shabnam Swarna, Dr. Sharifunnessa Moonmoon.

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
