## [Decision Letter · Decision Letter 0]

7 May 2025

Dear Dr. Moonmoon,

We look forward to receiving your revised manuscript.

Kind regards,

Ramegowda Venkategowda, PhD

Academic Editor

PLOS ONE

Journal Requirements:

“Ministry of The Science and Technology, Government of the Peoples Republic of Bangladesh [grant numbers -307; 2023-24, supported this work. “

5. Please provide a complete Data Availability Statement in the submission form, ensuring you include all necessary access information or a reason for why you are unable to make your data freely accessible. If your research concerns only data provided within your submission, please write "All data are in the manuscript and/or supporting information files" as your Data Availability Statement.

7. Please ensure that you refer to Figure 19 and 20 in your text as, if accepted, production will need this reference to link the reader to the figure.

Reviewers' comments:

Reviewer's Responses to Questions

**Comments to the Author**

1. Is the manuscript technically sound, and do the data support the conclusions?

Reviewer #1: Partly

Reviewer #2: Yes

Reviewer #3: Yes

Reviewer #4: No

2. Has the statistical analysis been performed appropriately and rigorously?

Reviewer #1: Yes

Reviewer #2: Yes

Reviewer #3: Yes

Reviewer #4: Yes

3. Have the authors made all data underlying the findings in their manuscript fully available?

Reviewer #1: Yes

Reviewer #2: Yes

Reviewer #3: Yes

Reviewer #4: Yes

4. Is the manuscript presented in an intelligible fashion and written in standard English?

Reviewer #1: No

Reviewer #2: Yes

Reviewer #3: Yes

Reviewer #4: No

Reviewer #1: Dear authors,

In this work, it has been studied the effects of exogenous application of different compounds on strawberries during field study. A detailed analysis of photosynthetic parameters is provided, and evidence that the treatments contribute for reducing the leaf temperature differential. The research is relevant in the context of alleviating heat stress, which is expected to increase due to climate change.

However, there are some major points that should be addressed. Certain aspects of the methodological setup require further clarification. I also provide a list of minor points for consideration. I see potential in the data acquired by the authors, and I hope my comments help to improve their work.

Kind regards

Major points

• English writing requires deep revision.

• Please provide the precise identification (variety) of the tested genotypes, as research reproducibility is fundamental for its acceptance. This information is also very relevant for future studies as, for instance, aiming at understanding why G1 is tolerant, while G2 and G3 are susceptible to heat stress.

• I think that is necessary a better description of how the G and P lines are related (sections 2.3 and 2.6). I suggest the authors to create a schematic representation.

• In the context of the comment above, it was also not clear for me which variety was used for the treatments. For instance, in Figures 6-18, three varieties are represented on panels b, d and f (P1-3). Do they correspond somehow with the genotype used for the treatments test (panels a, c, and e)?

• Please provide the full details of soil fertilization (concentration and manufacturer, amount and source of the manure, etc), fungicide and pesticide (manufacturer, concentrations).

• Please also mention at which time of the day, and at which day of the month the measurements were made. This is important for understanding why some of the observed effects did not appear on the first month of the analysis.

• The manuscript would benefit from a deeper bibliography input. That goes from the decision for the selection of the compounds for treatments to the analysis of the results. Statements in the Results should be supported by a source reference, even if the information is taken for granted. The impact of heat stress on the photosynthetic apparatus has been greatly studied in other species, often in combination with biotic or chemical stimulants. The deeper bibliographic review would also help to improve the Discussion, which has only 5 cited references.

Minor aspects

• Regarding the experimental setup (point 2.2), the authors mention that the treatments were performed during constant intervals over three months and repeated thrice, between October-November 2023 to April-May 2024. Next (point 2.7), I understand that an initial set was prepared and from this set, the three experiment rounds were started in parallel. However, this point is elusive and I do suggest to clarify it. Was there a time window between the start of each experimental round? If not, I would consider these not as three independent experiments, but rather as one larger experiment with a good number of replicates (Figure 3). While this design can be valuable for statistical analysis, especially in a field experiment, I would still recommend repeating the study to confirm the results.

• Scientific names should be italicizes (e.g. Fragaria, Rosaceae)

• Some abbreviations are lacking their full description next to its first mention (e.g. RCBD, PGR; RH);

• Please provide the reference for the RCBD model.

• I suggest to move the description of treatments that are now in the section 2.7 to section 2.8, so it would be next to the full description of the treatment application. Also, please provide the manufacturer of each reagent used. I would also suggest to clearly specify the temperature at which the reagents were stored;

• All Figures would benefit from better legends. Please introduce more details, including full description of what they are representing, description of the error bars, statistical test applied, number of individuals tested per condition (n), etc.

• Please also provide the version of each software and R package used.

• I suggest that each treatment be described by both its full name and abbreviation (T1–T5) when presenting the results. This would make the reading easier. Also, please add the description of each treatment o the figures’ legend.

• I suggest to increase the font size of Figure 19 for easier reading. Also, although not really critical, blue and red are usually used for denoting negative and positive values/correlation, so I suggest to swap them to avoid any misinterpretation. I understand that data across all genotypes and treatments were combined and used to plot the Pearson’s correlation. This information should be clear. Furthermore, I do suggest to review the data analysis and interpretation, as for instance the observation that “SPAD increases ambient temperature”, which is unlikely. Still, I wonder whether it might be more informative to represent the correlations of the tested parameters separately for each genotype and treatment. This way one would be able to evaluate their inputs independently.

Reviewer #2: Comments to the author (PONE-D-25-14863)

The manuscript titled “Mitigation of heat stress using hormones during summer season for survivability of strawberry plants” is an interesting study. The study by Salwa et al is mainly focusing on the use of five exogenous substances (Molasses, Melatonin, Kaolin, CaCl₂, and Abscisic Acid) to mitigate heat stress in three strawberry genotypes under tropical summer conditions which involves physiological and fluorescence measurements across different months. To be honest, the study is interesting and I truly enjoy the reading of manuscript. Overall, the research is valuable and relevant, particularly in the context of climate resilience in horticultural crops. However, some extensive improvements are needed to enhance the scientific clarity, rigor, and writing quality. As per my observation, in the current form, it lacks hypothesis-driven design clarity, and some methodological and interpretational issues need attention. Some suggestions are as follow:

It is extremely embarrassing to review the paper without line numbers to point-out the specific corrections. But I have following suggestions to consider:

I would strongly suggest to change the title as: Exogenous Application of Hormones and Chemicals to Mitigate Heat Stress in Strawberry Under Tropical Summer Conditions.

Overall, the abstract should be carefully revised as it lacks clarity and scientific soundness. Terms like “some sort of capability” are vague and has incomplete phrasing. Please provide better summary of the most effective treatments and genotype response needed.

In the introduction section, it needs better framing of research gaps. For example, what is unknown about hormone-based mitigation strategies in strawberry under field-like summer stress particularly in Bangladesh. Further, inconsistent tenses and awkward phrasing are present throughout, e.g. “It is establishing climatic change...” is an unclear statement. Further, “…mandatorily has to endure high heat” is overly informal. In Introduction, it doesn’t clearly state the hypothesis and research question as well as key gaps.

Material and methods are the most important section and it should have clear citations/protocol used in the present study. There are some key parts to improve. In this section, genotypes are inconsistently labeled, as sometimes G1/G2/G3, elsewhere P1/P2/P3 there must be uniformity throughout the manuscript. Please use scientific or cultivar names of strawberry genotypes if available and specify the concentration basis for each treatment (e.g., ppm or % w/v, and how these concentrations were finalized/selected. Please provide justification why these 5 treatments were chosen or provide appropriate literature citations/physiological rationales.

It is unclear that how many number of replicates per treatment/per genotype were used. Was the fluorometer standardized/calibrated before use? Any internal checks? Please elaborate the statistical models in more depth. Was random effect for replication included and interaction terms analyzed?

In the results section, many figures show overlapping trends please consider consolidating. Leaf surface temperature being higher in treatment than control contradicts expectation. Explanation is speculative; check methodology or measurement error. Graph captions needed improvements as some are mislabeled (e.g., gH shown but figure caption mentions ϕNPQ Fig. 18).

Please provide numerical values with statistical summaries (e.g., mean ± SE) in text for major parameters.

In statistical analysis part, please clarify what variables were considered random vs fixed in LMM. Further, Pearson correlation matrix (Fig. 19) is dense but informative and interpret more in text.

Overall discussion is descriptive, not mechanistic. Please include some information on the roles of ABA in stomatal regulation under heat, Melatonin as antioxidant/protective compound, Kaolin and molasses as physical shields against radiation, and also genotype differences are needed to be discussed. Please provide recent literature and add more context from global strawberry stress physiology.

Conclusion Needs to state clear practical recommendations, e.g., which combination(s) are best for growers. It must be concise to the point. Also, clarify that this is a first step, and in future long-term, field-based, and reproductive/yield trait studies are needed. add these as future directions.

Reviewer #3: Comments for Author

This manuscript “mitigation of heat stress using hormones during summer season for survivability of strawberry plants” is very interesting in which Author focused on effect of heat stress on strawberry and its alleviation through Hormones.

In abstract Author did not mention the levels or concentrations of hormones used to alleviate heat stress.

Authors required to write novelty statement about their research

Introduction is insufficient, Author should compare their research with studies of previous researchers

Hypothesis of study is not clear in introduction section

In material and methods section too much headings are used by Authors, it is required to write under minimum headings. Particularly second and third heading of the material and methods.

Why Author used RCBD design? Whereas experiment was conducted under pots.

Results are written well by authors but still revision is required to improve results. Author should add significance level or p values in results section.

Graphical presentation also required to improve. Authors need to write in correct manners on axis of graphs.

Pearson’s correlation diagram is not clear. It is advised to add clear diagram

In discussion section Author no need to indicate treatments as T1, T2.. etc Moreover, Author needs to write discussion in well mechanized form.

Conclusion section needs improvement and required to write limitation and future prospects of their study.

References should be balanced in reference list as well as in manuscript text.

Reviewer #4: In the present manuscript, the authors evaluated the effect of different PGRs sprays on three different strawberry genotypes to observe the tolerance response of plants from the onset of tropical summer to peak summer. The experimental approach is well justified and well described; however, the results are not adequately presented. In the result sections 3.2, 3.3 (number repeated in this section) and 3.4, there is no indication of which phenotype the data for the different treatments are (Figures, a, c, & e) or from which treatment the data in the phenotype graphs are (Figures, b, d & f). All of the above does not allow for understanding the interpretation of the results or their discussion.

Other minor concerns are:

- All abbreviations must be defined in the abstract and when they are first used in the introduction or materials and methods sections (RCBD, RH, SPAD, PGRs…).

- P1, P2 & P3 must be G1, G2 & G3 in section 2.6 of MM

- In the result sections 3.2, 3.3 (number repeated in this section) and 3.4, there is no indication of which phenotype the data for the different treatments are (Figures, a, c, & e) or from which treatment the data in the phenotype graphs are (Figures, b, d & f).

- In the MM section, three phenotypes are indicated: G1 (heat tolerant), G2, and G3 (heat susceptible). However, all the corresponding graphics are labeled with the P1, P2, and P3 varieties. Please clarify. These labels are repeated in all the results sections and in the discussion or conclusion sections.

Finally, I recommend that a native English speaker revise the entire manuscript…

**Do you want your identity to be public for this peer review?** For information about this choice, including consent withdrawal, please see our Privacy Policy

Reviewer #1: **Yes: ** Gustavo Turqueto Duarte

Reviewer #2: No

Reviewer #3: No

Reviewer #4: No

---

## [Author Response · Author response to Decision Letter 1]

9 Jun 2025

Author Responses to the Reviewer’s Comments

Section-A : latest part

Editor comments:

In your Methods section, please provide additional information regarding the permits you obtained for the work. Please ensure you have included the full name of the authority that approved the field site access and, if no permits were required, a brief statement explaining why.

Author responses:

No such body is exist at both institutional & country level and no such permission is needed or applicable for conduction of any research work or experiments regarding ethical considerations of plant culture and growth

This was added in section 2.1

Editor comments:

In the online submission form, you indicated that

"The data underlying the results presented in the study are available from corresponding author on request"

3. Uploaded as supplementary information.

Author responses

All data generated or analysed during this study are included in this manuscript. No more data, other than raw data, which also provided as supplementary file (3 excel file). Pls check declaration section and supplementary file (excel file)

Editor comments:

We note that your author list was updated during the revision process. In order to add or remove authors or update the order of the author byline after initial submission, we ask that authors complete an Authorship Change Request form. You may review our full authorship change policy and download the Authorship Change Request form here: https://journals.plos.org/plosone/s/authorship#loc-authorship-changes.

Please fill out all required sections of the form. If you are adding or removing more than 2 authors, you can complete multiple forms and submit as many forms as needed to reflect the updates. Please return the form(s) as an attachment by emailing plosone@plos.org or by uploading it as a submission file labeled with the file type ‘Other’. The form will be reviewed by PLOS staff for approval. Should the changes be approved, we will require written confirmation from all authors confirming that they approve of the changes. Please note that if your manuscript is accepted, we will not be able to complete the publication process without the completed form.

Author responses

This research was conducted under the direct supervision of the co-authors (Prof. Dr. Sharifunnessa Moonmoon & Prof. Dr. Md. Masudur Rahman - where they played role as Supervisor and co-supervisor respectively of the MS research of the first author (Najra-Tan-Nayeem Salwa). Mistakenly we excluded this co-author name

Now added even all the related formalities were completed and a humble request would like to made to include this co-author name

Authorship Change Request form was downloaded, filled up and uploaded in the system

This co-author name were included in everywhere of the related section (Pls see author name , affiliations , author contribution section)

Editor comments:

Please amend your authorship list in your manuscript file to include author Dr. Md. Masudur Rahman.

Author responses

This co-author name were included in everywhere of the related section (Pls see author name , affiliations , author contribution section)

Editor comments:

Please amend your manuscript to include a reference list. References must be placed at the end of the manuscript and numbered in the order that they appear in the text. For more information on the formatting of references, please visit the author guidelines at: http://journals.plos.org/plosone/s/submission-guidelines#loc-reference-style

Author responses

Amendment of the reference list was completed. References were placed at the end of the manuscript and numbered in the order that they appear in the text

Section –B : previous part

Reviewer #1: Dear authors,

In this work, it has been studied the effects of exogenous application of different compounds on strawberries during field study. A detailed analysis of photosynthetic parameters is provided, and evidence that the treatments contribute for reducing the leaf temperature differential. The research is relevant in the context of alleviating heat stress, which is expected to increase due to climate change.

However, there are some major points that should be addressed. Certain aspects of the methodological setup require further clarification. I also provide a list of minor points for consideration. I see potential in the data acquired by the authors, and I hope my comments help to improve their work.

Kind regards

Major points

Comments for Author:

English writing requires deep revision.

• Author’s Response

Thank you for your observation. We recognize that certain parts of the manuscript required improvement in English language and expression. To address this, we have thoroughly revised the manuscript to enhance clarity, grammar, and overall readability.

Comments for Author:

Please provide the precise identification (variety) of the tested genotypes, as research reproducibility is fundamental for its acceptance. This information is also very relevant for future studies as, for instance, aiming at understanding why G1 is tolerant, while G2 and G3 are susceptible to heat stress.

• Author’s Response

We fully agree that providing the precise identification of the tested genotypes is essential for ensuring reproducibility and facilitating future research. In response, we have specified the exact varieties used in the study: G1= RABI-3, G2= BARI Strawberry-2 and G3= BARI Strawberry-3. This information has been added to the Materials and Methods section (Section 2.2).

• Comments for Author:

I think that is necessary a better description of how the G and P lines are related (sections 2.3 and 2.6). I suggest the authors to create a schematic representation.

• Author’s Response

We understand the confusion caused by the inconsistent use of labels in Sections 2.3 and 2.6. Specifically, we mistakenly used “P” instead of “G” in some instances when referring to the genotypes. We have now carefully corrected this throughout the manuscript to consistently use “G” (i.e., G1 G2 and G3) for genotypes.

• Comments for Author:

In the context of the comment above, it was also not clear for me which variety was used for the treatments. For instance, in Figures 6-18, three varieties are represented on panels b, d and f (P1-3). Do they correspond somehow with the genotype used for the treatments test (panels a, c, and e)?

• Author’s Response

We appreciate your attention to the figure layout and labelling. To clarify, the treatments were indeed applied to all three genotypes (G1, G2 and G3). Figures 6-18 were designed to present two separate aspects of the study for each parameter: The effects of genotypes (shown in panels a, c and e) under various treatments and the overall effect of treatments (shown in panels b, d and f) averaged across genotypes.

Thus, both treatment and genotype effects were assessed within the same experimental setup.

Additionally, we acknowledge the labelling error in the genotype panels, where the genotypes were mistakenly labeled as P1, P2 and P3. These have now corrected to G1, G2 and G3 in the figures, figure legends, and relevant sections of the manuscript for clarity and consist

• Comments for Author:

Please provide the full details of soil fertilization (concentration and manufacturer, amount and source of the manure, etc), fungicide and pesticide (manufacturer, concentrations).

• Author’s Response

We agree that providing detailed information on soil fertilization, as well as the fungicide and pesticide applications, is important for transparency and reproducibility. In response, we have revised the Materials and Methods section (section 2.4) to include the following details:

The soil was mixed thoroughly with fertilizers following the BARC (2012) recommendation. Per pot, 250g vermicompost, 1.67 g TSP, 1.83 g MoP and 0.08 g MgSO4 were applied and kept for some days after watering to settle the soil.

• Comments for Author:

Please also mention at which time of the day, and at which day of the month the measurements were made. This is important for understanding why some of the observed effects did not appear on the first month of the analysis.

• Author’s Response

As mentioned in the revised manuscript, chlorophyll fluorescence data were collected during the first half of the day on three specific dates: 30th March, 30th April and 30th May. These points were strategically chosen to capture plant response under progressively increasing heat stress conditions, thereby allowing us to compare physiological performance across varying temperature intensities over time. We have now clearly stated this information in the Materials and Methods section (section 2.7).

Comments for Author:

The manuscript would benefit from a deeper bibliography input. That goes from the decision for the selection of the compounds for treatments to the analysis of the results. Statements in the Results should be supported by a source reference, even if the information is taken for granted. The impact of heat stress on the photosynthetic apparatus has been greatly studied in other species, often in combination with biotic or chemical stimulants. The deeper bibliographic review would also help to improve the Discussion, which has only 5 cited references.

• Author’s Response

Thank you for your insightful suggestion. We fully agree that strengthening the manuscript with a deeper and more comprehensive bibliographic foundation is essential to support our methodological choices and to place our findings within the context of existing research.

In response, we have significantly expanded the literature review in both the Introduction and Discussion sections.

Minor aspects

• Comments for Author:

Regarding the experimental setup (point 2.2), the authors mention that the treatments were performed during constant intervals over three months and repeated thrice, between October-November 2023 to April-May 2024. Next (point 2.7), I understand that an initial set was prepared and from this set, the three experiment rounds were started in parallel. However, this point is elusive and I do suggest to clarify it. Was there a time window between the start of each experimental round? If not, I would consider these not as three independent experiments, but rather as one larger experiment with a good number of replicates (Figure 3). While this design can be valuable for statistical analysis, especially in a field experiment, I would still recommend repeating the study to confirm the results.

• Author’s Response

To clarify, the three replications mentioned in the manuscript refer to biological replicates arranged simultaneously under the same environmental conditions—i.e., all genotypes and treatments were applied in parallel within a single experimental cycle. There was no time window between the start of each replication. Therefore, we agree that the setup represents one complete experiment with multiple replicates, rather than three temporally independent experimental rounds.

• Comments for Author:

Scientific names should be italicizes (e.g. Fragaria, Rosaceae)

• Author’s Response

We have reviewed and corrected the formatting of all scientific names, including genus and family names, to ensure they are properly italicized in accordance with scientific writing conventions (Pls see first line of Introduction).

Comments for Author:

Some abbreviations are lacking their full description next to its first mention (e.g. RCBD, PGR; RH-);

• Author’s Response

We have carefully revised the manuscript to ensure that all abbreviation, including RCBD, PGR and RH are now presented with their full forms upon first mention (Pls see ABSTRACT). This correction has been applied consistently throughout the text.

• Comments for Author:

Please provide the reference for the RCBD model.

• Author’s Response

We have now included a reference to support the use of the Completely Randomized Design (CRD) model applied in this experiment. The reference added is:

Gomez, K.A., & Gomez, A.A. (1984). Statistical Procedures for Agricultural Research (2nd ed.). John Wiley & Sons.

Comments for Author:

I suggest to move the description of treatments that are now in the section 2.7 to section 2.8, so it would be next to the full description of the treatment application. Also, please provide the manufacturer of each reagent used. I would also suggest to clearly specify the temperature at which the reagents were stored;

• Author’s Response

The description of treatments are now in the section 2.5 Specifications of the reagents with stored temperatures were provided (Pls see section 2.5)

Comments for Author:

All Figures would benefit from better legends. Please introduce more details, including full description of what they are representing, description of the error bars, statistical test applied, number of individuals tested per condition (n), etc.

• Author Response

All the recorded data were transferred in Microsoft Excel and data curating, sorting, arranging, processing, visualization etc., were performed. Data was then loaded in the R software (R Core Team, 2024) and RStudio ( Version: 2025.05.0+496) to carry out all kinds of statistical analysis. Linear mixed modelling was used to perform ANOVA in the light-dependent parameters. The light-independent parameters were analyzed by performing two-way ANOVA. The mean of the significant parameters were separated using the fisher LSD test (Gomez and Gomez, 1984) at 0.05 level of p-value. The “tidyverse” package was used for the calculation of mean and sem. The data was visualized in graph form using the “ggplot2” package along with others. Pearson’s correlation test were implied in the parameters using the “metan” package along with performing the significant test (Pls see section 2.8).

• Comments for Author:

Please also provide the version of each software and R package used.

• Author’s Response

In response, we have updated the Materials and Methods section to include the version details of all software and R packages used in the analysis. Specifically, statistical analyses were performed using R software version: 2025.05.0+496 (Pls see section 2.8).

• Comments for Author:

I suggest that each treatment be described by both its full name and abbreviation (T1–T5) when presenting the results. This would make the reading easier. Also, please add the description of each treatment o the figures’ legend.

• Author’s Response

We have revised the manuscript to present each treatment with both its full name and corresponding abbreviation (T1–T5) for clarity and ease of reading in result section. This includes a clear description of each treatment—5% Molasses (T1), 10 ppm Melatonin (T2), 5% Kaolin (T3), 10 mM CaCl₂ (T4), and 5 ppm Abscisic Acid (T5)—in the text.

• Comments for Author:

I suggest to increase the font size of Figure 19 for easier reading. Also, although not really critical, blue and red are usually used for denoting negative and positive values/correlation, so I suggest to swap them to avoid any misinterpretation. I understand that data across all genotypes and treatments were combined and used to plot the Pearson’s correlation. This information should be clear. Furthermore, I do suggest to review the data analysis and interpretation, as for instance the observation that “SPAD increases ambient temperature”, which is unlikely. Still, I wonder whether it might be more informative to represent the correlations of the tested parameters separately for each genotype and treatment. This way one would be able to evaluate their inputs independently.

• Author’s

---

## [Decision Letter · Decision Letter 1]

14 Jul 2025

Dear Dr. Moonmoon,

Thank you for submitting your manuscript to PLOS ONE. After careful consideration, we feel that it has merit but does not fully meet PLOS ONE’s publication criteria as it currently stands. Therefore, we invite you to submit a revised version of the manuscript that addresses the points raised during the review process.

We look forward to receiving your revised manuscript.

Kind regards,

Ramegowda Venkategowda, PhD

Academic Editor

PLOS ONE

Journal Requirements:

Additional Editor Comments:

Dear Authors,

Kindly sees the comments from a reviewer especially Reviwer 4. There is a serious issue on labeling the figures and given proper legends or methodological explanation. Kindly correct the same.

Reviewers' comments:

Reviewer's Responses to Questions

**Comments to the Author**

Reviewer #3: All comments have been addressed

Reviewer #4: (No Response)

2. Is the manuscript technically sound, and do the data support the conclusions?

Reviewer #3: Yes

Reviewer #4: Partly

3. Has the statistical analysis been performed appropriately and rigorously?

Reviewer #3: Yes

Reviewer #4: Yes

4. Have the authors made all data underlying the findings in their manuscript fully available?

Reviewer #3: Yes

Reviewer #4: Yes

5. Is the manuscript presented in an intelligible fashion and written in standard English?

Reviewer #3: Yes

Reviewer #4: Yes

Reviewer #3: Author has addressed all the comments properly with reasons. Therefore my decision is accepted for further publication process

Reviewer #4: The authors have greatly improved the manuscript's language but some revisión is still needed. However, my concerns remain regarding the presentation of results as explained in the first review. The graphs in Figures 6 through 18 (the number in Figure 17 is repeated) represent the different parameters recorded (Ambient temperature on the leaf surface, Leaf internal temperature, Leaf thickness, SPAD, etc., for each treatment (T0-T5)). The primary concern is that authors do not specify whether the data in the graphs in sections a, c, and e of each figure correspond to all plants or a specific variety. Furthermore, the graphs in sections b, d, and f of each figure, which refer to the three varieties used, do not specify which treatment (T0-T5) the data are from. The authors should present the data collected for each Strawberry variety and each treatment used to allow for better interpretation, comparison, and discussion of the results.

**Do you want your identity to be public for this peer review?** For information about this choice, including consent withdrawal, please see our Privacy Policy

Reviewer #3: No

Reviewer #4: **Yes: ** Homero Reyes de la Cruz

---

## [Author Response · Author response to Decision Letter 2]

25 Jul 2025

Author Responses to the Additional Editors Comments

[Dear Sir,

Sorry, I tried several time but failed to build PDF appropriately. I observed that, during PDF formation / build figure -18 is converting figure 17, seems to be repetition, though content is is not changing

Therefore total figure number is changing in to 19, instead of 20]

Would like to request to consider the fact or technical error ]

1. Additional editor comments:

Kindly see the comments from a reviewer especially Reviewer 4. There is a serious issue on labeling the figures and given proper legends or methodological explanation. Kindly correct the same.

Author Responses:

We sincerely thank Reviewer 4 for the valuable feedback. We acknowledge the concern regarding figure labeling and the lack of proper legends or methodological explanations. In response, we have thoroughly revised the manuscript to ensure that:

•All figures are correctly labeled with consistent numbering and appropriate captions.

•Figure legends now provide clear and sufficient detail to be understood independently.

Author Responses to the Reviewer’s Comments

Reviewer #4:

1.Reviewer’s Comments

The authors have greatly improved the manuscript's language but some revision is still needed.

Author’s Response

We have carefully reviewed the entire text once again and made further revisions to enhance clarity, grammar, and overall readability. All changes have been highlighted in the revised version of the manuscript for ease of reference.

2. Reviewer’s Comments

However, my concerns remain regarding the presentation of results as explained in the first review.

Author’s Response

The previous comments of reviewer-4 regarding results in the first review was like below

“The experimental approach is well justified and well described; however, the results are not adequately presented. In the result sections 3.2, 3.3 (number repeated in this section) and 3.4, there is no indication of which phenotype the data for the different treatments are (Figures, a, c, & e) or from which treatment the data in the phenotype graphs are (Figures, b, d & f). All of the above does not allow for understanding the interpretation of the results or their discussion”.

Thanks reviewer for above valuable suggestions

In results (section 3.2 and 3.3) repetition was avoided carefully. In 3.2 the ambient temperature of the leaf surface (Fig. 6), leaf internal temperature (Fig. 7) and the differential temperature (Fig. 8) were described in control and treatment groups of the study

The findings were described properly with clarity. Pls see the color changed text (red color) in section 3.2 (line number 335-342; 344-348; 355-357)

In section 3.3, two important leaf attributes like leaf thickness (Fig. 9) and leaf greenness (SPAD) (Fig. 10) were stated clearly. Please see color changed text (red color) where important findings were highlighted (line number 373 – 377; 386 – 391)

In section 3.4, figure denotes the treatment variation of maximum quantum efficiency where, in March ABA showed the best performance but with the time of increasing temperature (April and May) Kaolin (T3) treatment showed the better performance. In case of figure b. d, f, G1 showed the maximum quantum efficiency compared to the other genotypes (Pls see the line 395- 399 ; 399 -401)

3. Reviewer’s Comments

The graphs in Figures 6 through 18 (the number in Figure 17 is repeated) represent the different parameters recorded (Ambient temperature on the leaf surface, Leaf internal temperature, Leaf thickness, SPAD, etc., for each treatment (T0-T5)). The primary concern is that authors do not specify whether the data in the graphs in sections a, c, and e of each figure correspond to all plants or a specific variety.

Author’s Response

Thanks. Repetition of figure 17 is avoided

Yes, now specified

Figure 17: Steady-State Proton Flux (vH) of PSII of Strawberry plants under different treatments

and genotypes. Treatment and genotypes having different letter varies significantly at

p ≤ 0.05

and

Figure 18: Proton conductivity (gH) of PSII of Strawberry plants under different treatment and

genotypes. Treatment and genotypes having different letter varies significantly at

p ≤ 0.05

The data in the graphs in sections (3.2 & 3.3) a, c, and e of each figure (8, 9) correspond to all plants. Pls see color changed text (line number 335 and 367)

4. Reviewer’s Comments

Furthermore, the graphs in sections b, d, and f of each figure, which refer to the three varieties used, do not specify which treatment (T0-T5) the data are from. The authors should present the data collected for each Strawberry variety and each treatment used to allow for better interpretation, comparison, and discussion of the results.

Author’s Response

To clarify, all treatments were indeed applied across all three genotypes (G1, G2, and G3). Figures 6–18 were intentionally designed to illustrate two complementary perspectives for each parameter:

• Panels a, c, and e show the effects of different genotypes under various treatments.

• Panels b, d, and f show the overall effects of treatments averaged across genotypes.

In other words, both genotype and treatment effects were analyzed within the same experimental setup. To generate these visualizations, we created two separate pivot tables from the raw data for each parameter—one summarizing data by genotype and the other summarizing data by treatment.

---

## [Editor Report · Decision Letter 2]

17 Aug 2025

Exogenous application of hormones and chemicals to mitigate heat stress in Strawberry under tropical summer conditions

PONE-D-25-14863R2

Dear Dr. Moonmoon,

We’re pleased to inform you that your manuscript has been judged scientifically suitable for publication and will be formally accepted for publication once it meets all outstanding technical requirements.

Kind regards,

Ramegowda Venkategowda, PhD

Academic Editor

PLOS ONE
---

## [Editor Report · Acceptance letter]

PONE-D-25-14863R2

PLOS ONE

Dear Dr. Moonmoon,

I'm pleased to inform you that your manuscript has been deemed suitable for publication in PLOS ONE. Congratulations! Your manuscript is now being handed over to our production team.

Kind regards,

on behalf of

Dr. Ramegowda Venkategowda

Academic Editor

PLOS ONE